# Systematic characterization of the HOXA9 downstream targets in MLL-r leukemia by noncoding CRISPR screens

Shaela Wright[1,7], Xujie Zhao [2,6,7], Wojciech Rosikiewicz [3], Shelby Mryncza[4], Judith Hyle[1], Wenjie Qi[3], Zhenling Liu[1], Siqi Yi[5], Yong Cheng [5], Beisi Xu [3] & Chunliang Li [1] ✉

Accumulating evidence indicates that HOXA9 dysregulation is necessary and sufficient for leukemic transformation and maintenance. However, it remains largely unknown how HOXA9, as a homeobox transcriptional factor, binds to noncoding regulatory sequences and controls the downstream genes. Here, we conduct dropout CRISPR screens against 229 HOXA9-bound peaks identified by ChIP-seq. Integrative data analysis identifies reproducible noncoding hits, including those located in the distal enhancer of *FLT3* and intron of *CDK6*. The Cas9-editing and dCas9-KRAB silencing of the HOXA9-bound sites significantly reduce corresponding gene transcription and impair cell proliferation in vitro, and in vivo by transplantation into NSG female mice. In addition, RNA-seq, Q-PCR analysis, chromatin accessibility change, and chromatin conformation evaluation uncover the noncoding regulation mechanism of HOXA9 and its functional downstream genes. In summary, our work improves our understanding of how HOXA9-associated transcription programs reconstruct the regulatory network specifying MLL-r dependency.

As a homeodomain transcription factor, HOXA9, together with 38 other proteins encoded by members of the HOX gene family, plays an essential role in the embryonic development of vertebrates by controlling body specification and anterior-posterior pattern formation temporally and spatially[1–3]. In normal hematopoiesis, HOXA9 expression is high in hematopoietic stem and progenitor cells (HSPC), whereas it is downregulated during differentiation[4–6]. A deficiency in HOXA9 led to a striking reduction in myeloid progenitors, granulocytes/monocytes precursors, and lymphoid precursors in mice[7]. In contrast, overexpression of HOXA9 resulted in enhanced proliferation of HSPCs[8]. Collectively, these results highlight the critical role of HOXA9 in regulating hematopoiesis, especially myeloid differentiation.

The clinical correlation identified between high HOXA9 expression and poor outcomes suggests the oncogenic activity of HOXA9[9,10]. Notably, overexpression of HOXA9 has been identified in several specific leukemia subtypes, including the MLL-r, NPM1c, EZH2-mutant, and NUP98-fusion subtypes. Experimental validation confirmed abnormally high expressions of the HOXA9 gene and its cofactor, MEIS1, in more than 70% of acute myeloid leukemia (AML) cases[11]. Co-expression of HOXA9 and MEIS1 is sufficient to induce leukemogenesis[12]. Similarly, ectopic expression of NUP98-HOXA9 in mouse hematopoietic stem cells led to AML transformation with a long latency, which could be accelerated with the concomitant expression of MEIS1[12]. In AML cells, the oncogenic function of HOXA9 was exerted

[1]Department of Tumor Cell Biology, St. Jude Children's Research Hospital, 262 Danny Thomas Place, Memphis, TN 38105, USA. [2]Department of Pharmacy and Pharmaceutical Sciences, St. Jude Children's Research Hospital, 262 Danny Thomas Place, Memphis, TN 38105, USA. [3]Center for Applied Bioinformatics, St. Jude Children's Research Hospital, 262 Danny Thomas Place, Memphis, TN 38105, USA. [4]Department of Biology, Rhodes College, 2000 North Pkwy, Memphis, TN 38112, USA. [5]Department of Hematology, St. Jude Children's Research Hospital, 262 Danny Thomas Place, Memphis, TN 38105, USA. [6]Present address: Zhongda Hospital, School of Life Sciences and Technology, Advanced Institute for Life and Health, Southeast University, Nanjing 210096, China. [7]These authors contributed equally: Shaela Wright, Xujie Zhao. ✉e-mail: Chunliang.Li@stjude.org

by regulating cell proliferation and differentiation. Ablation of HOXA9 led to the arrested proliferation and enhanced apoptosis, especially in MLL-r cells[13]. Hence, HOXA9 represents a promising therapeutic target. However, targeting HOXA9 directly as a transcription factor is quite challenging. As an alternative, an epigenetic silencing upstream activator of HOXA9 has been investigated[14–17]. For instance, a large epigenetic complex consisting of the MLL-fusion protein and its co-factors, DOT1L[18], WDR5[19], MENIN[20], and LEDGF[21] maintains constitutive expression of HOXA9 by a direct binding at the HOXA9 locus; therefore, inhibitors that disrupt the interaction between the MLL-fusion protein and its co-factors represent a promising way of abolishing the HOXA9 expression. So far, inhibitors targeting DOT1L have been developed and evaluated in clinical trials even though they have demonstrated limited therapeutic efficacy when used as single agents[22]. The MENIN inhibitors targeting the interaction between MLL1 and MENIN have also shown promising therapeutic potential in preclinical evaluations[23,24]. We previously established an endogenous HOXA9-knockin reporter in an MLL-r B-ALL cell line, SEM[25]. We have identified positive and negative regulators responsible for HOXA9 transcription through a loss-of-function CRISPR screen[26]. In contrast, few studies have explored the direct and functional targets of HOXA9 to reveal the molecular mechanism of its downstream signaling as an avenue to pursue potential therapeutic targets.

In this work, we use a CRISPR-mediated loss-of-function screen against HOXA9-bound peaks identified by ChIP-seq to systematically evaluate HOXA9-bound *cis*-regulatory elements in MLL-r leukemia cell models. We also conduct integrative analyses to reveal the long-range chromatin interaction regulation mechanism of genes essential for survival through HOXA9/MEIS1 binding occupancy. We demonstrate the downstream functional effectors of HOXA9 and provide additional clues to identify potential therapeutic targets in MLL-r leukemia.

## Results

### Genome-wide CRISPR dropout screening reveals essential HOXA9-bound *cis*-regulatory elements critical for leukemia maintenance

Considering the transcription factor HOXA9 is an oncogene and critical to the survival of MLL-r leukemia cells, exploring its direct downstream targets promises to provide insights into the HOXA9 regulome and to identify additional therapeutic targets. We hypothesized that genome editing-mediated disruption of HOXA9-bound sites should mimic the loss-of-function effect of HOXA9. This approach would affect functional target gene expression, impair cell fitness, and prevent functional compensation by other *HOXA* genes. To this end, we used Cas9-mediated CRISPR screening to perturb HOXA9-bound sites genome-wide. Remarkably, no high-quality published ChIP-seq data is available to locate reproducible HOXA9-bound sites. Based on these observations, we established a doxycycline-modulated Tet-On system to ectopically express HA-tagged HOXA9 cDNA in MLL-r SEM cells (Fig. 1A and Supplementary Fig. 1A). Then, ChIP-seq was performed using specific antibodies against the HA tag in both no-doxycycline and doxycycline-treated cells. In total, we identified 229 reproducible HOXA9-bound peaks in SEM cells. Among these peaks, 50% bind to introns, 12.5% bind to promoters, and 29.1% bind to distal regions (Fig. 1B). Motif analysis confirmed that the top consensus motif is HOXA9 (Fig. 1C). Next, we analyzed the co-factor profiling of the 229 peaks based on collected ChIP-seq data of transcription factors (TFs) from human B-ALL SEM. These data suggest MEIS1 (a known co-factor of HOXA9) and other leukemia-specific TFs co-bind with HOXA9 (Fig. 1D and Supplementary Fig. 1B).

Next, we designed 5718 sgRNAs to target the 229 HOXA9-binding peaks, including 100 non-target sgRNAs and 120 positive-control sgRNAs against ribosomal genes. The pooled lentiviral sgRNA library was used to infect Cas9-expressing or dCas9-KRAB-expressing MLL-r SEM cells, followed by a survival-based dropout screen. The loss of

representation of sgRNAs at day 7 and day 14, as compared with their presence on day 0, implies that CRISPR targeting of the candidate targeted regions impairs cell growth and survival, as exemplified by sgRNAs targeting *RPS19*, an essential gene that served as a positive control (Fig. 1E–I). The MAGeCK[27] analysis algorithm was used to identify sgRNA depleted during the two-week culture. Notably, sgRNAs (FLT3[Enh]-left and right) targeting a known distal enhancer of the *FLT3* locus[28] were among the top drop-out candidate sgRNAs (FDR < 0.05) in both Cas9 and dCas9-KRAB screens (Fig. 1F–I). Our dropout screen system against HOXA9-bound noncoding regions reliably identified putative target genes of the HOXA9 protein.

### Epigenetic regulation of *FLT3* transcription and the biological significance of *HOXA9* and *FLT3* regulation through the distal enhancer regulation

The *FLT3* gene is well known to be essential for survival in MLL-r subtypes based on genomic and genetic evidence[29–31]. Although several studies reported the HOXA9 protein could bind and regulate *FLT3* at the promoter in mouse settings[32,33], the detailed transcriptional regulation mechanism has not been addressed. Moreover, we identified a different HOXA9-bound site distant from *FLT3* in MLL-r leukemia cells. To further explore the regulatory mechanism of the HOXA9-bound *cis*-regulatory element regulating the *FLT3* gene, we used integrative analysis of epigenetic profiling and genome editing approaches. First, we analyzed publicly available ATAC-seq datasets derived from a panel of human acute lymphoblastic leukemia (ALL) cell lines with and without MLL rearrangement and 13 normal human hematopoietic lineages[34]. A distal human *cis*-regulatory element (CRE) (h-FLT3) approximately 170 kb upstream of *FLT3* showed significantly higher chromatin accessibility in MLL-r ALL cells (SEM and RS4;11 cells), as compared with non-MLL-r ALL cells (697, Nalm6, REH, SUPB15, and UOCB-1 cells) (Fig. 2A). Interestingly, a solid ATAC-seq peak was also observed in HSPCs (CD34 + , HSC, MPP, CMP, GMP, MEP, and CLP cells) but not in mature lineages (monocytes, erythrocytes, CD4 + , and CD8 + T cells, NK cells, and B cells)(Fig. 2A)[34]. Similarly, the enhancer activity of h-FLT3 was supported by more robust H3K27ac ChIP-seq signals in MLL-r acute myeloid leukemia (AML) cells (MOLM13, MOLM14, and MV4;11 cells) as compared with non-MLL-r AML cells (IMSM2, HEL, K562, SET2, U937, and HL60 cells) and normal CD34+ cells (Fig. 2B)[35–37]. Notably, within the *FLT3* distal enhancer, we confirmed a specific binding peak of both HOXA9 and MEIS1 using our HOXA9-HA ChIP-seq and publicly available HOXA9 and MEIS1 ChIP-seq datasets. Notably, the HOXA9-bound peaks did not show CTCF binding occupancy within the ± 2Kb, suggesting that transcription regulation through HOXA9 likely occurs independently of co-regulation from the looping factor CTCF.

By exploring HiC data from SEM cells[25], we found evidence of a long-range DNA interaction between the *FLT3* promoter and the HOXA9/MEIS1-bound h-FLT3 enhancer (Fig. 2C). Promoter Capture-C data further confirmed this looping event (Fig. 2D). We have previously established an auxin-inducible degron model of looping factor CTCF in SEM cells[25], allowing acute depletion of CTCF protein (Fig. 2E). Using this cellular model, we demonstrated that upon CTCF loss, FLT3 expression was indeed not altered at both protein and total mRNA levels (Fig. 2F–H). In addition, we also confirmed that nascent RNA of FLT3 and neighbor gene PAN3 were not changed as characterized by SLAM-seq (Supplementary Fig. 2A, B). Interestingly, neither the chromatin accessibility at the h-FLT3 enhancer nor the chromatin loop between the *FLT3* promoter and enhancer was altered (Fig. 2I), suggesting that additional chromatin looping factors other than CTCF may be involved in controlling *FLT3* transcription.

Consistent with previous reports[29–31,38–40], *FLT3* expression was significantly higher in MLL-r leukemia cell lines ($n = 11$) than in non-MLL-r leukemia cell lines ($n = 89$) ($p < 0.0001$). Also, MLL-r leukemia cell lines ($n = 8$) showed more survival dependency on *FLT3* than non-

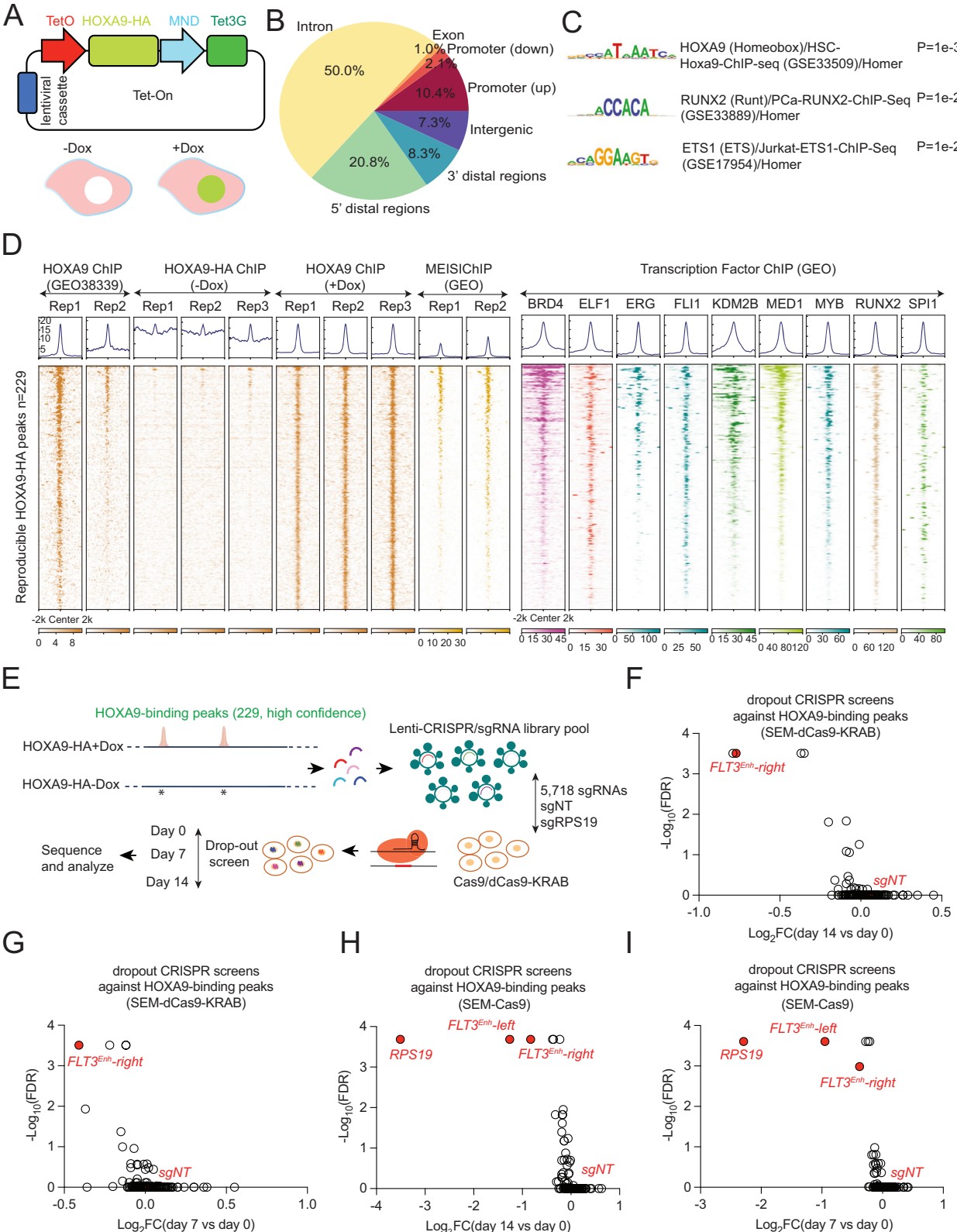

MLL-r leukemia cell lines ($n = 39$) ($p < 0.0001$), according to the CRISPR essential score from the DepMap dataset[41]. Furthermore, there was a significant negative correlation ($r = -0.684$, $p = 8.28e{-}8$) between the CRISPR essential score and *FLT3* expression (Supplementary Fig. 3A–C). Of note, according to the TCGA database, high *FLT3* expression was associated with poor overall survival in a cohort of 106

patients with AML (Supplementary Fig. 3D, E). A similar trend was also observed in the TARGET and BEAT cohorts (Supplementary Fig. 3F–I). Collectively, the results of the epigenetic profiling, the chromatin interaction, and the expression correlation suggest that there is h-FLT3-mediated transcriptional activation of *FLT3* in MLL-r leukemia cells.

**Fig. 1 | A CRISPR editing screen discovered HOXA9 binding sites essential for the growth of MLL-r leukemia cells. A** Diagram of the all-in-one inducible Tet-On system to ectopically express HOXA9-HA cDNA. **B** Genomic distribution of the 229 HOXA9 ChIP-seq peaks. **C** The top three consensus motifs and *P* value were shown using the HOMER motif analysis algorithm against 229 peaks compared with control peaks. **D** Heat maps of HOXA9-HA ChIP-seq peaks compared with publicly available HOXA9 and MEIS1 ChIP-seq datasets and additional transcription factor ChIP-seq datasets in SEM cells (GEO: GSE117864)[70]. **E** The 229 reproducible HOXA9-bound peaks were identified from our ChIP-seq data and targeted by a library of 5718 sgRNAs, including 100 non-target (NT) sgRNAs used as negative controls and

120 positive control sgRNAs, respectively. Through genome editing mediated by Cas9, drop-out CRISPR library screens were performed to select HOXA9-bound sites critical for the survival of SEM cells on day 7 and day 14. **F–I** Volcano plot of enriched sgRNAs from the drop-out screen using SEM Cas9 and dCas9-KRAB cells infected with the sgRNA library and collected at day 7 and day 14. The positive control sgRNAs targeting the coding region of the RPS19 gene were enriched at all Cas9-mediated screens. The sgRNAs targeting an *FLT3* enhancer (left site peak and right site peak) were enriched in all screens. FDR was calculated based on the MAGeCK algorithm descrbed in the Methods.

## Characterization of the HOXA9/FLT3 regulation phenotype and mechanism in MLL-r leukemia

Inspired by the profiling and correlation analysis results, we conducted functional validation in both MLL-r and non-MLL-r cell lines. We found that the HOXA9-bound site is specific at the distal enhancer of FLT3 but not the promoter (Fig. 3A, B). Therefore, two sgRNAs (sgFLT3-DE-1 and sgFLT3-DE-2) from the CRISPR sgRNA library were selected to disrupt the HOXA9 occupancy motif at the h-FLT3 enhancer (Fig. 3C). TIDE-seq analysis confirmed that more than 90% indel frequency was observed (Supplementary Fig. 4A, B). As a result, Cas9-mediated genome editing by these two sgRNAs resulted in significant downregulation of *FLT3* expression and the adjacent *PAN3*, compared with that of the non-target control (Fig. 3D). Subsequently, we conducted a time-course competitive proliferation assay (CPA) to characterize the impact on the proliferation of decreased *FLT3* expression. The CFP fluorescence in the sgRNA-expressing lentiviral vectors was used to monitor the compromised cell survival observed in MLL-r SEM cells (Fig. 3E), as targeted by sgFLT3-DE-1 and sgFLT3-DE-2. Similar results were observed in two other MLL-r AML cell lines (MOLM13 and MV4;11) (Fig. 3F–G) but not in non-MLL-r cell lines (Nalm6 and GM12878) (Fig. 3H, I). Of note, the MLL-r AML OCI-AML2 cells expressed high *FLT3* expression but displayed weak h-FLT3 enhancer activity as indicated by H3K27ac ChIP-seq (Fig. 3C). Therefore, as expected, there was no change in cellular phenotype in OCI-AML2 cells infected with sgFLT3-DE-1 and sgFLT3-DE-2 (Fig. 3J). In summary, genome editing of the HOXA9 binding site in the h-FLT3 enhancer functionally impaired cell proliferation in the context of MLL-r.

It has been reported that double-stranded breaks may induce cell death and may cause the HOXA9-binding motif-independent phenotype[42,43]. To mitigate this potential impact, we designed additional complementary experiments to confirm the result obtained from Cas9-mediated genome editing. Firstly, a full-length wildtype *FLT3* cDNA was overexpressed via the lentiviral system in the h-FLT3 enhancer-targeted SEM cells. This resulted in a significant rescue at the protein level and consistent recovery in CPA (Supplementary Fig. 4C, D). Second, we used the CRISPR interference (CRISPRi) assay by blocking the chromatin accessibility of the HOXA9-bound site at the h-FLT3 enhancer with the same sgRNA DE-1 in SEM cells stably expressing dCas9-KRAB (Fig. 4A). As expected, on a genome-wide scale, the ATAC-seq peak that was most significantly decreased upon CRISPRi targeting by sgFLT3-DE-1, as compared with the non-target control sgRNA, corresponded to our target locus (FDR < 0.05) (Fig. 4B, C). Subsequently, expressions of both *FLT3* and *PAN3* were significantly downregulated compared with the non-target control sgRNA (Fig. 4D, E). Moreover, consistent with our observations with Cas9-edited cells, cell growth was significantly retarded, as indicated by the time-dependent decrease in the CFP+ percentage of dCas9-KRAB SEM cells that were positive for sgFLT3-DE-1 (Fig. 4F). However, the interaction between the *FLT3* promoter and the h-FLT3 enhancer was not altered by blocking the binding of HOXA9 at the h-FLT3 enhancer with sgFLT3-DE1 sgRNA, as indicated by Capture-C using two baits (bait 1 and 2) against the *FLT3* promoter (Fig. 4G). Our previous ChIP-seq data showed that the binding peaks of CTCF and

HOXA9 at this locus are ~5Kb away. Therefore, targeting the HOXA9-bound site does not affect CTCF or CTCF-associated chromatin interaction. Also, it is possible that other chromatin looping factors are required for *FLT3* enhancer/promoter regulation. Finally, *FLT3* was an essential gene in MLL-r according to the DepMap dataset (Fig. 4H). As expected, MLL-r SEM cells exhibited superior sensitivity to pharmacological inhibition of FLT3 by Gilteritinib (LC50: 54 nM) when compared to Nalm6 (LC50: 1852 nM), REH (LC50: 1523 nM), and 697 (LC50: 1729) cells (Fig. 4I). Moreover, sgFLT3-DE-1-mediated CRISPRi blunted the drug response of SEM cells to Gilteritinib (Fig. 4J).

To test further the effect of disrupting the h-FLT3 enhancer on the in vivo growth of SEM cells, we performed an in vivo competitive assay by transplanting a mix of sgFLT3-DE1 SEM cells (CFP+ cells) expressing either dCas9-KRAB or Cas9 and parental SEM cells (CFP- cells) into NSG mice at a ratio of 4:1 (Fig. 5A). The percentage of CFP+ SEM cells out of the total SEM cells (hCD45+ cells) in the peripheral blood was monitored by flow cytometry. As shown in Fig. 5B, C, the in vivo cell growth of SEM cells positive for sgFLT3-DE-1 was outcompeted by parental SEM cells 2 and 3 weeks after injection. Our use of integrative genetic tools revealed the HOXA9-binding and *cis*-regulatory regulation mechanism of *FLT3* in MLL-r leukemia models.

## Functional validation of additional HOXA9 targets

In addition to sgRNAs (sgFLT3-DE-1 and sgFLT3-DE-2) targeting h-FLT3 enhancer, several other sgRNAs were also enriched in the dropout CRISPR screen. To test the functional relevance with MLL-r survival, we conducted a CPA assay in Cas9-expressing SEM and MOLM13 cells targeted with sgRNAs against the HOXA9-bound sites in the genes of *CDK6*, *RUNX1*, *DCAF11*, *PEBP4*, *AHI1*, *NDUFS8*, ZCCHC7 and *MAN1C1*. TIDE-seq analysis confirmed that indels were observed in the coding regions of each gene by CRISPR-Cas9 targeting (Supplementary Fig. 5). Consistent with our dropout screen results, all the targeted cells showed impaired proliferation (Fig. 6A, B). To independently validate the targeting effect, the sgRNAs against *CDK6*, *RUNX1*, and *DCAF11* were delivered into dCas9-KRAB expressing SEM and MOLM13 cells. CRISPRi system reproducibly confirmed the decreased proliferation phenotype (Fig. 6C, D). To further reveal the molecular mechanism underlying the HOXA9-target gene regulation, we focused on the *CDK6* locus. The HOXA9-bound site is located in the intronic region and co-localizes with enhancer markers H3K27ac and BRD4 (Fig. 1D and Supplementary Fig. 6). HiC data also reveal strong loops between this region and the *CDK6* promoter (Fig. 6E). We also conducted ATAC-seq on SEM cells targeted with sgCDK6 (HOXA9-bound site in *CDK6* intron) and sgNT. These data demonstrated that CRISPRi against the HOXA9-bound site in *CDK6* intron significantly decreased the chromatin accessibility and the *CDK6* expression (Fig. 6E–G, and Supplementary Fig. 7A, B). In contrast, chromatin accessibility of *DCAF11* and *RUNX1* loci was not obviously altered, pointing to the presence of alternative mechanisms (Supplementary Fig. 7C, D). Total RNA-seq and differential gene expression analysis confirmed that the *CDK6* transcription is the most significantly decreased gene upon CRISPRi (FDR < 0.01), further supporting distal noncoding regulation is specific to *CDK6* (Fig. 6H).

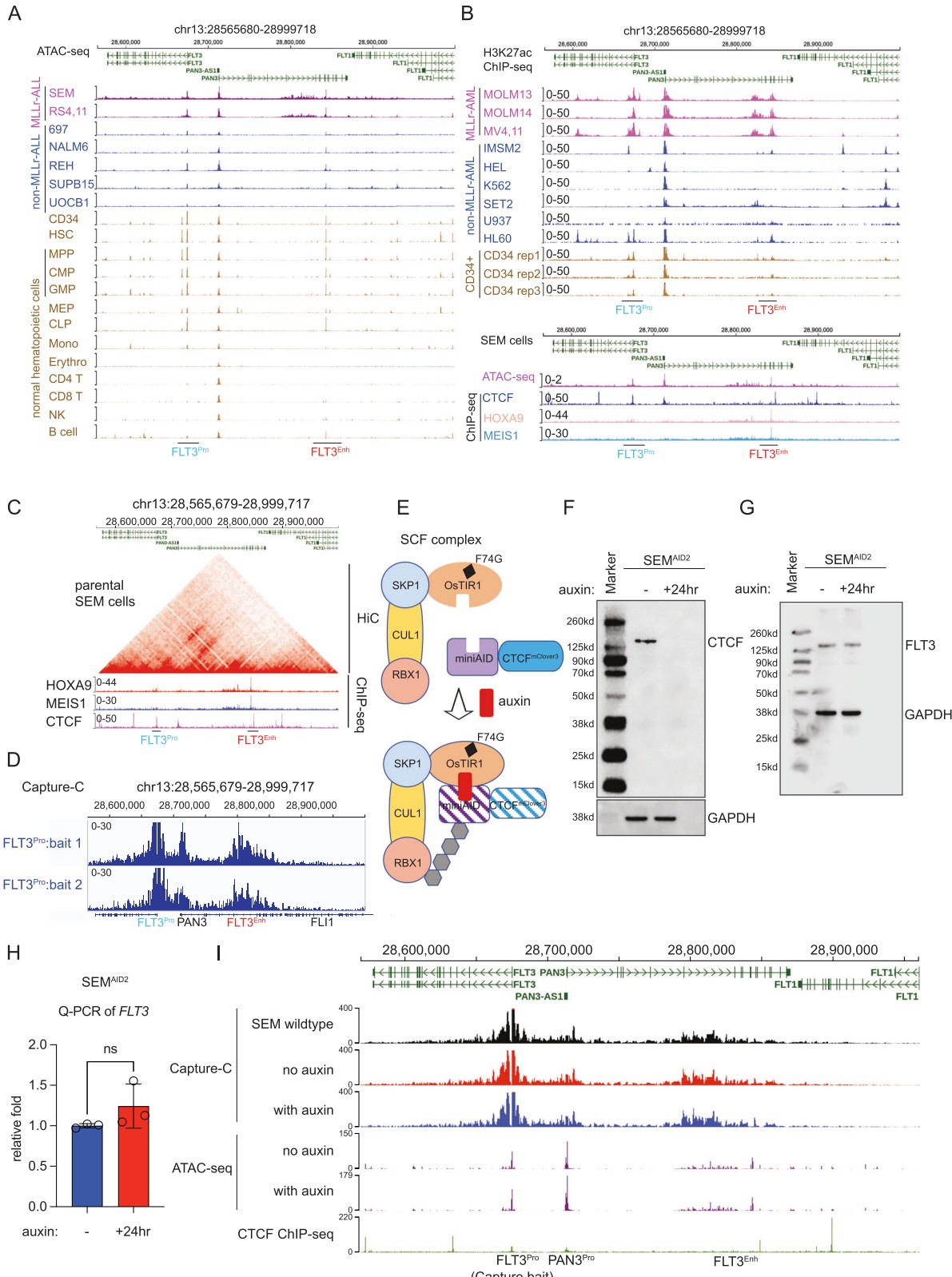

In addition to HOXA9-FLT3 and HOXA9-CDK6 regulation axis, it will be interesting to investigate the noncoding regulation of other downstream target genes by HOXA9 in a follow-up study in the future. Taken together, these extensive efforts will undoubtedly improve our understanding of how HOXA9 works in MLL-r leukemia maintenance and provide insights for alternative targeting and therapeutic innovation. Fig. 7

## Discussion

Numerous studies have suggested that dysregulation of HOXA9 is a dominant driver in the pathogenesis of MLL-r leukemia. Aberrant high expression of HOXA9 is a hallmark of MLL-r leukemia, including AML and ALL, and is associated with a poor prognosis for this disease. Nonetheless, the exact downstream targets of HOXA9 accounting for its oncogenic role remain to be fully elucidated due to the lack of

**Fig. 2 | Characterizing the epigenetic regulation of *FLT3*. A** Functional annotation of the *FLT3* locus, using ATAC-seq data from seven B-ALL leukemia cell lines either with the *MLL* gene rearrangement (MLL-r cells) (SEM and RS4;11) or without MLL rearrangement (non-MLL-r cells) (697, Nalm6, REH, SUP-B15, and UOC-B1) (GEO: GSE129066)[68] and from 13 different normal human hematopoietic lineages (GEO: GSE74912)[34]. A potential *FLT3* enhancer (FLT3^Enh) at -170 kb upstream of *FLT3* was observed in all leukemia cell lines, normal human stem cells, progenitor cells, and B cells. The signal of the *FLT3* enhancer was significantly more robust in cells with the *MLL* gene rearrangement than in cells without the *MLL* rearrangement. **B** A strong H3K27ac ChIP-seq signal was observed at the FLT3 enhancer in AML cells with the *MLL* rearrangement but not in AML cells without MLL rearrangement (upper panel)(GSE17312[71], GSE65138[36], GSE80779[35], GSE79899[72], GSE109492[73], GSE137652[74], GSE111179[76], GSE111293[75]). Strong CTCF, HOXA9, and MEIS1 ChIP-seq signals were also observed at the h-FLT3 enhancer locus in SEM cells (lower panel). **C** HiC data from parental SEM cells demonstrated long-range chromatin interaction between the *FLT3* promoter and the h-FLT3 enhancer (upper panel)(GEO: GSE138862). ChIP-seq data (HOXA9, MEIS1, and CTCF) served as a reference to locate the HOXA9-bound site (lower panel). **D** Capture-C was conducted to characterize the chromatin looping between the FLT3 promoter and the h-FLT3 enhancer using biotin-labeled DNA oligos against the FLT3 promoter. **E** Schematic diagram of auxin-inducible degron system to target and acutely deplete CTCF protein. **F** Immunoblotting was conducted to confirm the acute protein degradation of CTCF in the presence of auxin for 24 hours. The results were confirmed by consistent two replicates and one representative result was shown here. **G** Immunoblotting was performed to detect the expression of FLT3 protein upon acute degradation of CTCF. GAPDH was included as an internal control. **H** Q-PCR was used to examine the mRNA expression of *FLT3* upon CTCF protein degradation. Data are shown as mean values ± SEM of three biological replicates (center of the error bar). *P* values were estimated using a two-tail un-paired *t*-test. **I** Capture-C and ATAC-seq were conducted to characterize the chromatin accessibility change and chromatin conformation change following CTCF degradation, respectively. The biotin-labeled DNA oligos against the *FLT3* promoter were used as baits to quantify chromatin looping between the *FLT3* promoter and the h-FLT3 enhancer. Source data are provided as a Source Data File.

commercial ChIP-seq grade antibody available[44]. Moreover, HOXA9 is undruggable, adding another layer of difficulty to leverage the therapeutic potential of targeting HOXA9. CRISPR-based (either Cas9 or dCas9) screenings have recently been reported to efficiently manipulate the transcription of target genes by perturbating noncoding *cis*-regulatory elements[45–47]. Inspired by these studies, we established the HA-tag-based HOXA9 ChIP-seq using an inducible expression system and identified 229 reproducible HOXA9-bound peaks in MLL-r leukemia cell lines. Subsequently, we designed a library of sgRNAs targeting binding sites of HOXA9 and performed Cas9-based genome editing, aiming to determine functional *cis*-regulatory elements: 1) directly targeted by HOXA9; 2) responsible for cell fitness. Notably, several distal *cis*-regulatory elements influencing cell fitness associated with poorly explored target genes were identified. Targeting the HOXA9-bound sites by CRISPR/Cas9 or CRISPRi reduced the chromatin accessibility of targeted regions (Fig. 4B and Supplementary Fig. 7A, B), substantially compromised the enhancer activity of HOXA9-bound sites. However, targeting other HOXA9-bound sites at the promoters of *RUNX1* and *DCAF11* did not impact chromatin accessibility, indicating other mechanisms may exist (Supplementary Fig. 7C, D).

In this study, we explored the functional binding sites of HOXA9 using CRISPR/Cas9-based genome editing, and a subsequent dropout CRISPR screen in MLL-r positive SEM cells. We identified a series of HOXA9-driven *cis*-regulatory elements that are critical for the survival of MLL-r leukemia cells, including h-FLT3, a distal enhancer critical for the high expression of *FLT3* in MLL-r leukemia and its survival[33,48]. Another example is HOXA9-binding and regulation at the intronic region of *CDK6*. Both genes were essential for MLL-r leukemia cell survival. Our integrative mechanism studies revealed that chromatin-looping mediated enhancer represents another layer of regulation in the HOXA9/ downstream gene axis. Our data also suggest that additional looping factors other than CTCF may be required for the HOXA9 regulation.

Notably, most of the functionally validated HOXA9-bound sites co-localized with enhancer markers as H3K27ac and BRD4 regardless of their genomic location (Supplementary Fig. 6), suggesting HOXA9 may be functional through controlling enhancer activity. These observations are consistent with the previous report that HOXA9 may reprogram the enhancer landscape to promote leukemogenesis[49]. However, more stringent functional assays and research tools are required to explore the detailed mechanism.

High expression of *FLT3* has been recognized as a critical component of the unique identity of MLL-r ALL, compared to non-MLL-r ALL and AML[50]. However, the underlying mechanism for the dysregulation of *FLT3* expression is largely unknown. In this study, we have identified a distal enhancer (h-FLT3) highly responsible for *FLT3* transcription under the direct control of HOXA9. Disrupting the binding of HOXA9 at this enhancer via CRISPR editing resulted in significant downregulation

of *FLT3*. Collectively, our results determined FLT3 as a downstream effector of HOXA9 and established the axis of MLL-r fusion, *HOXA9*, and *FLT3* in MLL-r leukemia. Moreover, high *FLT3* expression also conferred preferential sensitivity to FLT3 inhibition, especially in MLL-r leukemia cells[38,39]. Consistent with these observations, down-regulation of *FLT3* by disrupting HOXA9 binding at the h-FLT3 enhancer led to decreased growth of MLL-r leukemia cells in vitro and in vivo and decreased their sensitivity to the FLT3 inhibitor, Gilteritinib.

HOXA9 overexpression was also identified in many different leukemia subtypes, including onco-fusion proteins, NPM1c+, etc.; experimental validation of the target binding site and cellular function is interesting for understanding HOXA9's role in other HOXA9-driven leukemia cells. Inspired by the NUP98-HOXA9 fusion containing the homeodomain of HOXA9 protein, we think it is also worth using the NUP98-HOXA9 AML model to further evaluate the HOXA9 binding peak function.

In summary, we have conducted a genome-wide CRISPR screen for HOXA9-binding sites, leading to a refined understanding of the role of HOXA9 in noncoding segment regulation. The system itself was confirmed to be a powerful tool for identifying downstream targets of HOXA9 that represent potential therapeutic targets for MLL-r and potentially for other HOXA9-driven leukemia (Fig. 7).

## Methods
Our research complies with all relevant ethical regulations. Protocols for mouse studies were approved by the St. Jude Children's Research Hospital Institutional Animal Care and Use Committee.

### Cell culture
The REH, Nalm6, and MV4;11 cell lines were originally purchased from ATCC, and the SEM, OCI-AML-2, and 697 cell lines were originally purchased from DSMZ. MOLM13 was originally purchased from ACCEGEN. GM12878 was from Coriell Institute. All cell lines were maintained in RPMI 1640 medium (GIBCO, Life Technologies) supplemented with 10% heat-inactivated fetal bovine serum and 2 mM L-glutamine, at 37 °C in 5% $CO_2$. A high-titer virus was generated using 293 T cells in this study, which were maintained in DMEM (GIBCO, Life Technologies) supplemented with 10% heat-inactivated fetal bovine serum and 2 mM L-glutamine, at 37 °C in 5% $CO_2$. All cell lines were validated by STR before use in this study. All cell lines used in this study were Mycoplasma-free examined by Lookout Mycoplasma PCR Detection Kit (Sigma, #MP0035) following the instruction.

### Vectors
The Lenti-Cas9-Blast (Addgene #83480), Lenti-dCas9-KRAB-blast (Addgene #89567), and the Lenti-Guide-Puro (Addgene #52963) plasmids were purchased from Addgene. The IRES-CFP cassette was

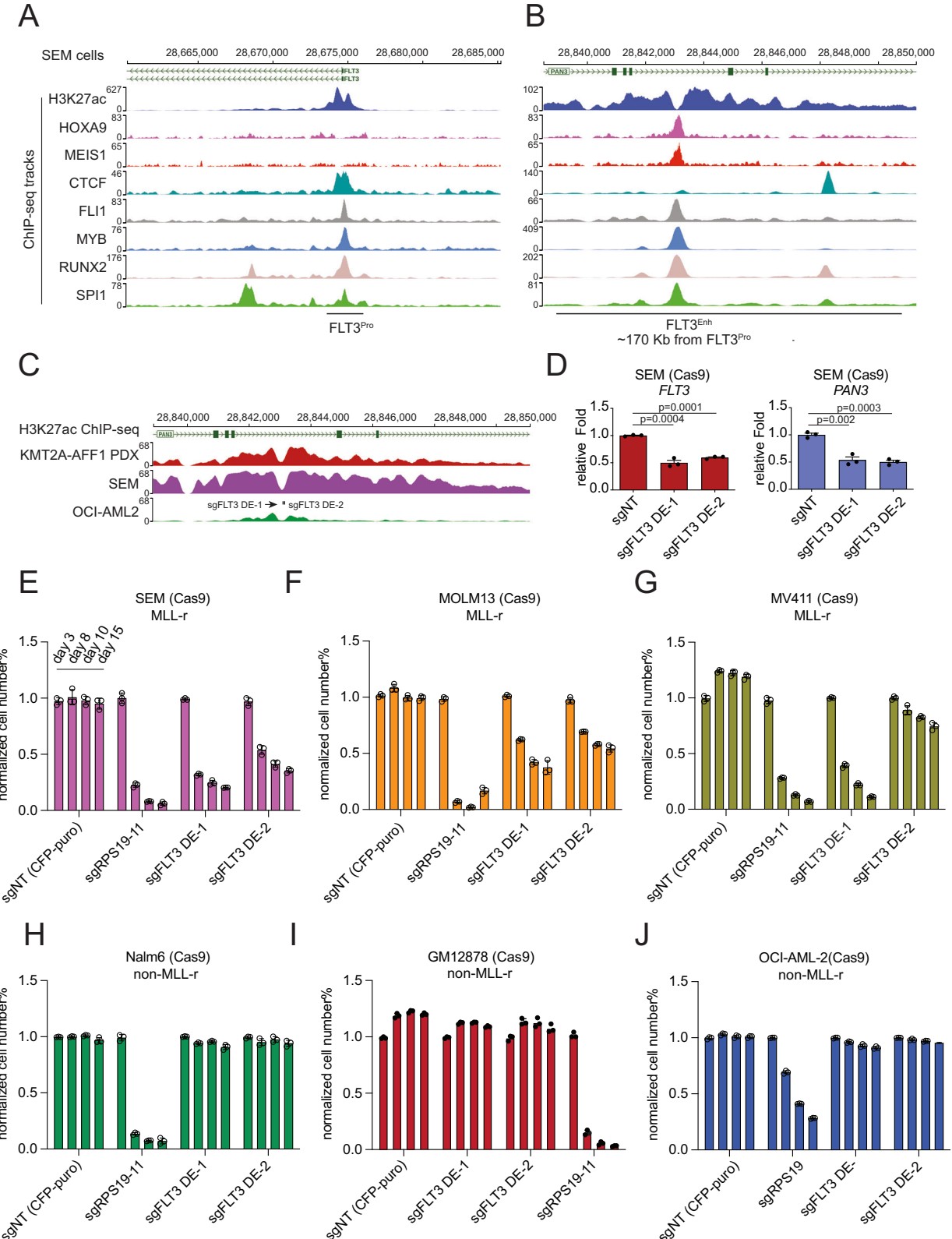

**Fig. 3 | Cas9-mediated disruption of the h-FLT3 enhancer blunted MLL-r leukemia cell growth.** ChIP-seq tracks of transcription factors and H3K27ac (GEO: GSE117864)[70] were used to demonstrate the epigenetic status of the *FLT3* promoter (**A**) and the HOXA9-bound site in the h-FLT3 enhancer (**B**). **C** Two sgRNAs (sgFLT3-DE-1 and -DE-2) targeting the h-FLT3 enhancer were used to disrupt the h-FLT3 enhancer activity. **D** Q-PCR detection of gene expression demonstrated notable downregulation of *FLT3* and *PAN3* expressions in SEM cells upon CRISPR targeting.

Cas9/sgFLT3-DE-1/2-mediated disruption of the h-FLT3 enhancer resulted in retarded cell growth of MLL-r leukemia cells (**E**–**G**) but not non-MLL-r leukemia cells or OCI-AML-2 cells with low enhancer activity of the h-FLT3 (**H**–**J**). The sgRPS19-11 sgRNA served as the positive control. The percentage of cell numbers was normalized to CFP+ control cells (sgNT). Data are shown as mean values ± SEM of three biological replicates (center of the error bar). *P* values were estimated using a two-tail un-paired *t*-test. Source data are provided as a Source Data File.

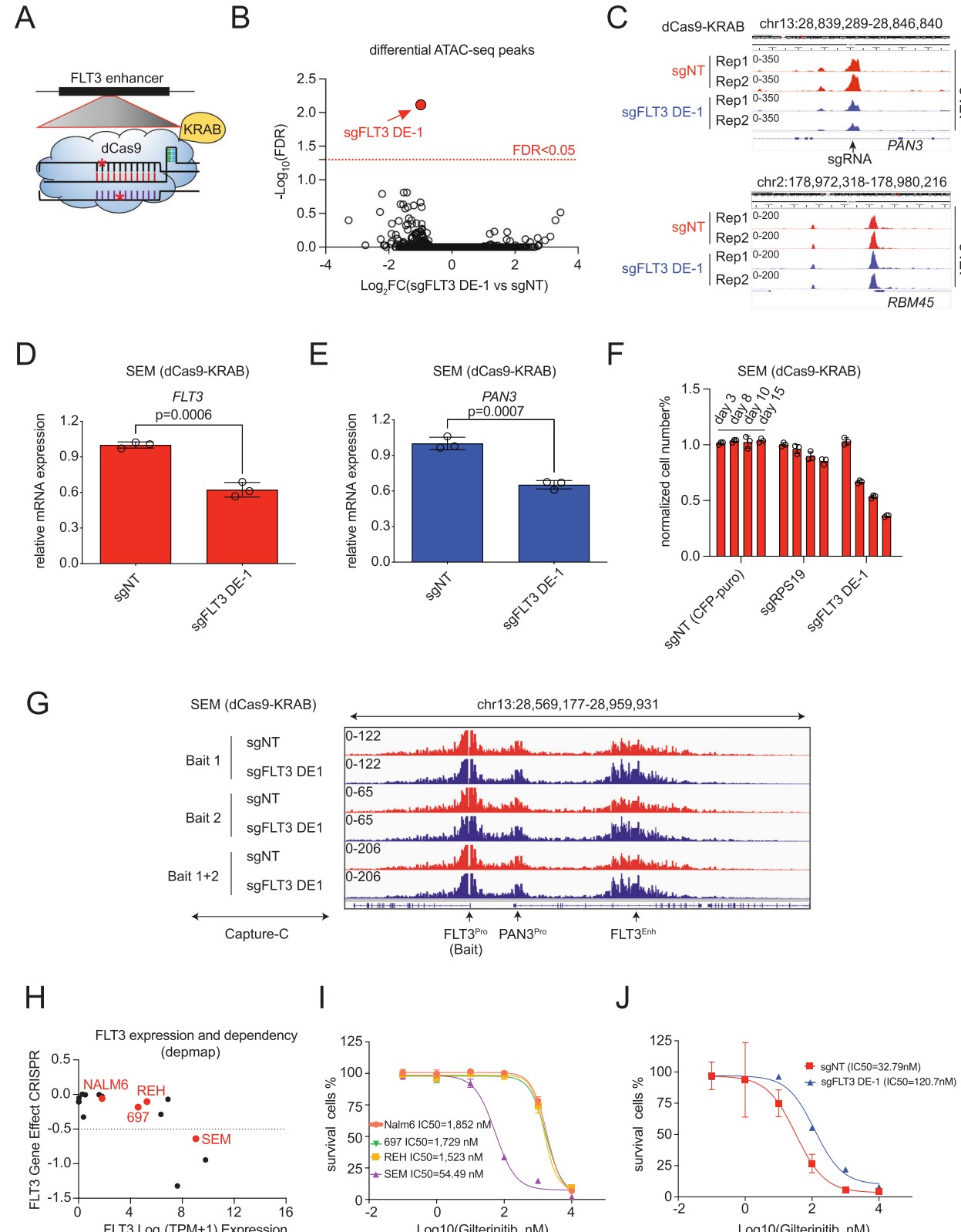

cloned into the Lenti-Guide-Puro plasmid to enable the flow tracing of sgRNA-targeted cells in a competitive proliferation assay. A pair of oligonucleotides containing a 20-bp sgRNA sequence targeting the candidate region was synthesized (Thermo Fisher Scientific) and cloned into the Lenti-Guide-Puro-IRES-CFP construct between two BsmBI sites. Correct clones were selected by screening and confirmed by Sanger sequencing with the U6-Forward sequencing primer (5'-

GAGGGCCTATTTCCCATGAT-3').For lentiviral overexpression of *FLT3*, the *FLT3* cDNA containing the coding region was cloned with a pair of primers (Supplementary Data 1) and inserted into the lentiviral expression vector by Gibson Assembly. SnapGene software was used to design all primers used for cloning. The PCR amplifications of products for cloning were performed using CloneAmp polymerase (Clontech), and the cycling parameters were as follows: 98 °C for

**Fig. 4 | dCas9-KRAB-mediated disruption of the h-FLT3 enhancer blunted growth of MLL-r leukemia cells and decreased sensitivity to an FLT3 inhibitor.** **A** A dCas9-KRAB-mediated CRISPR interfering system was applied to disrupt the h-FLT3 enhancer. **B**, **C** ATAC-seq results showed that Cas9-KRAB/sgFLT3-DE-1 only influenced the chromatin accessibility of the h-FLT3 enhancer locus in a genome-wide analysis. The non-relevant *RBM45* locus was one of the negative loci. **D**–**F** dCas9-KRAB/sgFLT3-DE-1-mediated disruption of the h-FLT3 enhancer led to down-regulation of *FLT3* and *PAN3* expressions and retarded cell growth of SEM cells. Expressions of FLT3 and PAN3 were normalized to SEM (dCas9-KRAB) cells infected with non-target sgRNA (sgNT). The percentage of cell numbers was normalized to CFP⁺ control cells. Data are shown as mean values ± SEM of three biological replicates (center of the error bar), and results represent three independent

experiments. *P* values were estimated using a two-tail un-paired *t*-test. **G** Capture-C results showed a strong interaction between the h-FLT3 enhancer and the *FLT3* promoter, and this interaction was not influenced by dCas9-KRAB/sgFLT3-DE-1-mediated disruption of the h-FLT3 enhancer. Two baits (Bait 1 and Bait 2) were designed against the *FLT3* promoter. *FLT3* was shown as an essential gene in SEM (MLL-r) cells, indicated by gene effect (<-0.5 is recognized as essential)(DEPMAP) (**H**). Its high expression conferred SEM superior sensitivity to the FLT3 inhibitor Gilteritinib (**I**), whereas dCas9-KRAB/sgFLT3-DE-1-mediated disruption of the h-FLT3 enhancer led to the down-regulated expression of *FLT3* and decreased sensitivity to Gilteritinib (**J**). Data are shown as mean % viability relative to vehicle ± SEM of three biological replicates (center of the error bar), and results represent three independent experiments. Source data are provided as a Source Data File.

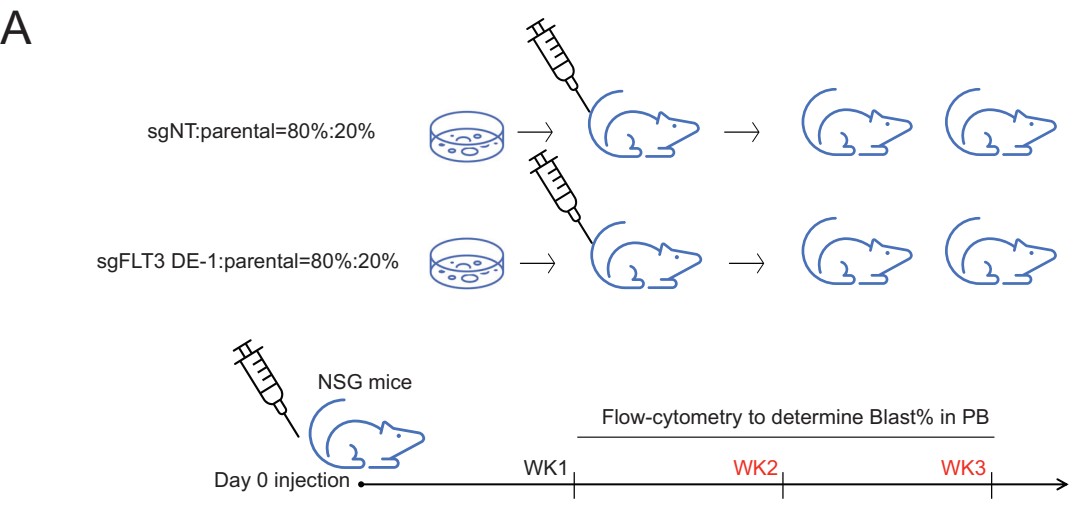

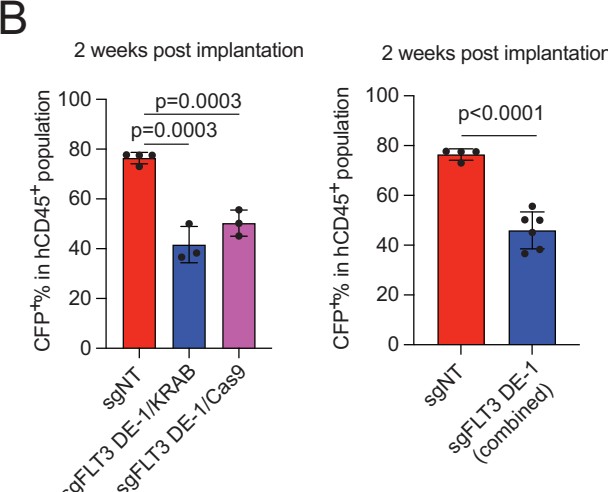

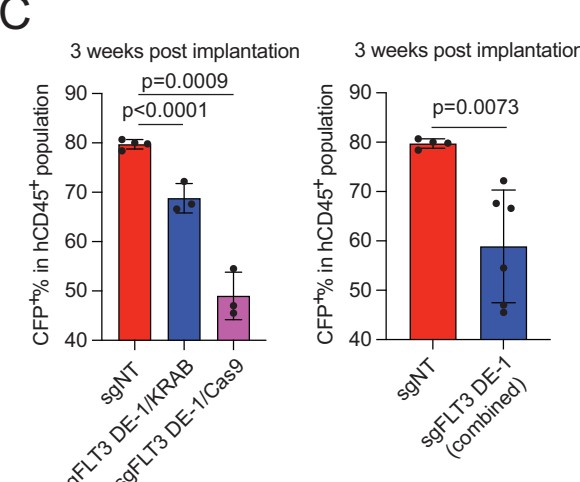

**Fig. 5 | CRISPR-based disruption of the h-FLT3 enhancer blunted in vivo MLL-r leukemia cell growth.** **A** A diagram of in vivo competitive transplantation in female NSG mice to evaluate the effect of CRISPR-based disruption of the h-FLT3 enhancer on cell growth of SEM cells. The percentage of SEM cells in the peripheral blood was monitored weekly by flow cytometry. CRISPR-based (dCas9-KRAB and Cas9) disruption of h-FLT3 enhancer compromised in vivo

growth of SEM cells at 2 weeks (**B**) and 3 weeks (**C**) after transplantation. Four animals in sgNT group and three animals in targeted experiment groups were used. Data are shown as mean values ± SEM of three biological replicates (center of the error bar). *P* values were estimated using a two-tail un-paired *t*-test. Source data are provided as a Source Data File.

5 min, followed by 40 cycles of 98 °C for 15 s, 55 °C for 20 s, and 72 °C for 20 s. A Gibson Assembly Cloning Kit (NEB #E5510S) was used in accordance with the manufacturer's instruction, and all reactions were carried out at 50 °C for 20 min. All primer information was included in Supplementary Data 1.

### CRISPR library construction and screening
A set of approximately 5718 sgRNA oligonucleotides that targeted 229 HOXA9 binding peaks carrying consensus motifs was designed for array-based oligonucleotide synthesis (CustomArray). The unique binding of each sgRNA was verified by sequence blast against the

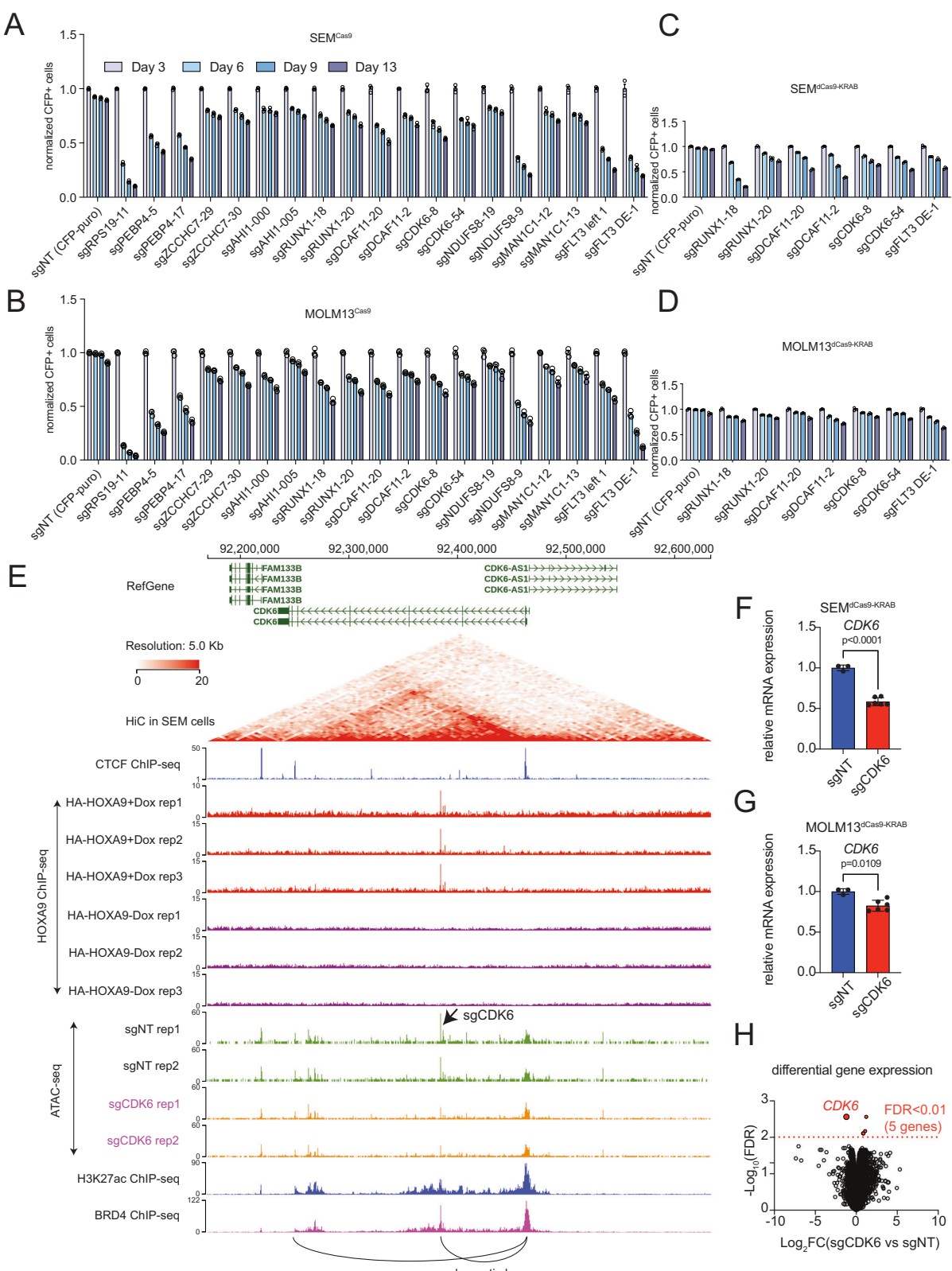

whole human genome in the sgRNA pooled library. The synthesized oligonucleotide pool was amplified by PCR and cloned into the LentiGuide-Puro-IRES-CFP backbone using an In-Fusion HD Cloning Kit (Clontech #638909). The Cas9-expressing MLL-r SEM cell line was infected with the pooled sgRNA library at a low M.O.I (approximately 0.3). Transduced cells were selected by blasticidin and puromycin. After the cells had recovered from the selection process, the pooled population was collected on day 0, and portions were maintained in liquid culture for 7 or 14 days. The sgRNA sequences were recovered by genomic PCR analysis and deep sequencing using a HiSeq 4000 or Novasesq system (Illumina) for single-end 150-bp reads. The primer sequences used for cloning and sequencing are listed in Supplementary Data 1. The sgRNA sequences and counts are described in Supplementary Data 2.

**Fig. 6 | Functional interrogation of HOXA9-bound targets by genome editing.**
**A** Competitive proliferation assay was conducted in Cas9-expressing SEM cells targeted with sgRNAs against HOXA9-bound sites close to the genes as *PEBP4, ZCCNC7, AHI1, RUNX1, DCAF11, CDK6, NDUFS8,* and *MAN1C1*. The sgRPS19 and sgFLT3-DE-1 sgRNA served as positive controls. Disruption of the loci targeted by these selected sgRNAs led to retarded cell growth of SEM cells in a time-dependent manner. The percentage of cell numbers was normalized to CFP⁺ control cells infected with non-target sgRNA (sgNT). **B** Competitive proliferation assay was conducted in Cas9-expressing MOLM13 cells targeted with sgRNAs against HOXA9-bound sites close to the genes as *PEBP4, ZCCHC7, AHI1, RUNX1, DCAF11, CDK6, NDUFS8* and *MAN1C1*. The sgRPS19 and sgFLT3-DE-1 sgRNA served as positive controls. Disruption of the loci targeted by these sgRNAs led to retarded cell growth of MOLM13 cells in a time-dependent manner. The percentage of cell numbers was normalized to CFP⁺ control cells infected with non-target sgRNA (sgNT). **C** Competitive proliferation assay was conducted in dCas9-KRAB-expressing SEM cells targeted with sgRNAs against HOXA9-bound sites close to the genes *RUNX1, DCAF11,* and *CDK6*. The sgFLT3-DE-1 sgRNA served as positive

control. **D** Competitive proliferation assay was conducted in dCas9-KRAB-expressing MOLM13 cells targeted with sgRNAs against HOXA9-bound sites close to the genes *RUNX1, DCAF11,* and *CDK6*. The sgFLT3-DE-1 sgRNA served as positive control. **E** Characterization of the chromatin conformation change upon dCas9-KRAB targeting against the HOXA9-bound site in the intron of *CDK6*. HiC (GEO: GSE138862), HOXA9 ChIP-seq, H3K27ac ChIP-seq and BRD4 ChIP-seq (GEO: GSE117864)[70] tracks were shown to characterize the epigenetic status of the HOXA9-bound site. **F** Q-PCR was conducted to quantify the transcription decrease of *CDK6* when CRISPRi targeted the HOXA9-bound site in the intron of CDK6 in SEM cells. **G** Q-PCR was conducted to quantify the transcription decrease of *CDK6* when CRISPRi targeted the HOXA9-bound site in the intron of CDK6 in MOLM13 cells. **H** Total RNA-seq was performed using SEM cells targeted with sgCDK6 against the HOXA9-bound site in the CDK6 intron. Differential gene expression was defined by FDR < 0.01. The *CDK6* expression is the top hit. Data are shown as mean values ± SEM of three biological replicates. *P* values were estimated using a two-tail unpaired *t*-test. Source data are provided as a Source Data File.

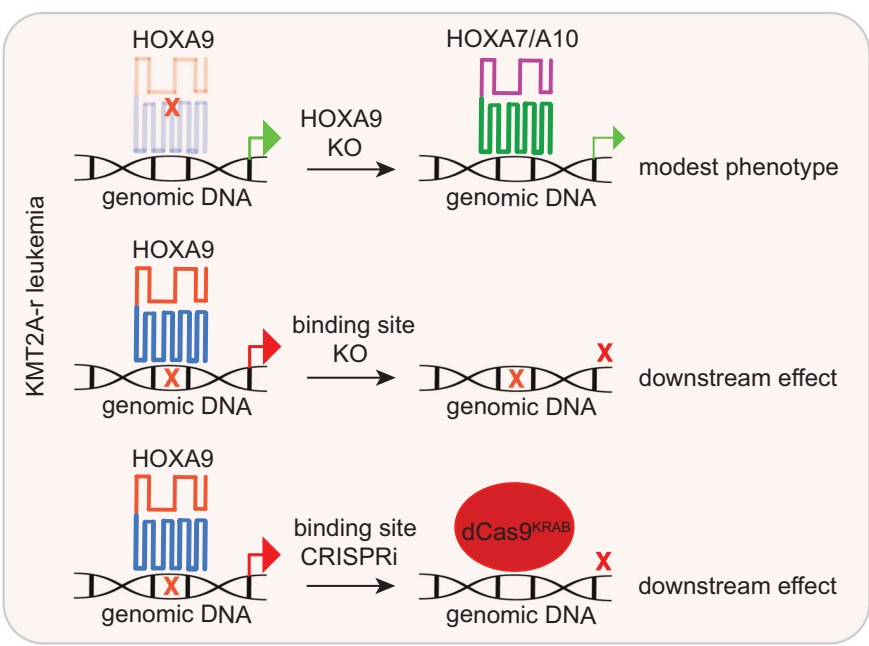

**Fig. 7 | Schematic diagram summarizing this study.** Our work interrogated the transcription factor function of HOXA9 in MLL-r leukemias by targeting HOXA9-bound sites by genomic editing (CRISPR/Cas9 and CRISPRi). It would overcome the common issue observed by HOXA9 protein perturbation and HOX compensation.

## Data analysis of CRISPR screening

Raw data 150-bp reads were obtained with an Illumina NovaSeq system and trimmed for adapters. The raw FASTQ data were de-barcoded and counted 20mers or 19mers by bbmap (version 37.28, "kmercountexact.sh fastadump = f mincount = 1 k = 20 rcomp = f") and assigned to sgRNAs. MAGeCK (version 0.5.9.4, default parameters) was used for statistical analysis of results. Detailed screening results were included in Supplementary Data 2.

## Quantitative real-time PCR

Total RNA was collected using TRIzol reagent (Thermo Fisher Scientific #15596026) or Direct-zol RNA Miniprep Kit (Zymo Research #R2052). Reverse transcription was performed using a High-Capacity cDNA Reverse Transcriptase Kit (Applied Biosystems #4374966). Real-time PCR was performed using FAST SYBR Green Master Mix (Applied Biosystems #4385612) in accordance with the manufacturer's instructions. Relative gene expression was determined by using the ΔΔ-CT method[51]. All Q-PCR primers used in this study are listed in Supplementary Data 1.

## RNA-seq

Total RNA was extracted by Trizol (Thermo Fisher Scientific #15596026) from replicate samples. About 200 ng total RNA was treated using Kapa rRNA depletion reagents to remove ribosomal RNA, then converted into cDNA libraries using Kapa RNA HyperPrep Kit with RiboErase (HMR). After the end repair procedure, dA-tailing, and adapter ligation, each cDNA library was purified and enriched by 11 cycles of PCR amplification.

## RNA-seq data analysis

Paired-end 11-cycle sequencing was performed on the NovaSeq 6000 sequencer following the manufacturer's instructions (Illumina). Raw reads were first trimmed using TrimGalore (v0.6.3) with parameters '--paired --retain_unpaired.' Filtered reads were then mapped to the Homo sapiens reference genome GRCh37.p13 using STAR (v2.7.9a)[52]. Gene-level read quantification was done using RSEM (v1.3.1)[53] on the Gencode annotation v19[54]. To identify the differentially expressed genes, normalization factors were first estimated using the TMM, and genes with CPM < 1 in all samples were removed. Next, the

TMM normalization factors and raw counts were then used for the Limma-voom analysis using the "voom," "lmFit" and "eBayes" functions from the limma R package[55]. Gene set enrichment analysis (GSEA) was performed using the MsigDB database (v7.1)[56,57]. Differentially expressed genes were ranked based on their $log_2$(FC). RNA expression value was provided in Supplementary Data 3.

## Competitive proliferation assay

To evaluate candidate sgRNAs' impact on leukemia expansion, cell cultures were transduced with individual sgRNAs in CFP expressing lentiviral vectors, followed by measurement of the percentage of CFP-positive cells at various days post-infection, using flow cytometry[26,58]. The CFP-positive percentage was normalized to that at the starting time point and declined over time. The decline was used to infer a defect in cell accumulation conferred by a given sgRNA associated with survival, relative to the uninfected cells in the same culture. An infected pool population could also be injected into host NSG mice via the tail vein. Blood samples collected from the eyeballs were used for flow cytometric analysis of the CFP-positive percentage.

## Immunoblotting

Cell lysates were prepared by using RIPA buffer. Lysates were subjected to electrophoresis on an SDS-PAGE gel (Thermo Fisher Scientific) and transferred to a PVDF membrane (Bio-rad) according to the manufacturer's protocols (Bio-Rad) at 100 V for 1 h. After a blocking incubation with 5% non-fat milk in TBS-T (10 mM Tris, pH 8.0, 150 mM NaCl, 0.5% Tween-20) for 1 h at room temperature, membranes were incubated with antibodies against GAPDH (Thermo Fisher Scientific #AM4300, 1:5,000), HA (Abcam #18181, 1:1,000), AID (MBL #M214-3, 1:2000), HOXA9 (Atlas Antibodies #HPA061982, 1:1000), FLT3 (Cell Signaling, #3462, 1:1,000) or CTCF (Santa Cruz, #sc-271514, 1:200) at 4 °C overnight with gentle shaking. Membranes were washed three times for 10 min each with TBS-T and incubated with dilution of sheep anti-mouse IgG HRP (GE Healthcare, NA931, 1:5000), or dilution of donkey anti-rabbit IgG HRP (GE Healthcare, NA340, 1:5000) for 1 h at room temperature. After washing with TBS-T three times for 10 mins each, the membranes were developed with the ECL system (Perkin Elmer) according to the manufacturer's protocol.

## ChIP-seq analysis

Twenty million HOXA9-HA cells were treated with 1ug/mL doxycycline for 48 hours to induce the HOXA9-HA protein expression. DMSO treatment was used as a negative control. Cells were fixed with 1% formaldehyde for 5 mins at room temperature (Covaris TruChIP Chromatin Shearing Kit). Nuclei were prepared according to the TruChIP protocol, and chromatin was sheared in a Covaris milli tube using the Covaris M220 ultrasonicator set at a duty factor of 10 and 200 cycles/burst for 10 min at set point 6 °C. Sheared chromatin was centrifuged for 10 mins at $8000 \times g$, and clarified chromatin was moved to a new 1.5-mL Eppendorf tube. Chromatin was amended to a final concentration of 50 mM Tris−HCL pH 7.4, 100 mM NaCl, 1 mM EDTA, 1% NP-40, 0.1% SDS, and 0.5% Na deoxycholate plus protease inhibitors (PI). About 60ul of washed anti-HA magnetic beads (Pierce, catalog #88837) were added to the chromatin overnight with rotation at 4 °C. The next day, samples were placed on a magnetic stand, unbound chromatin was removed, and beads were washed two times with wash buffer 1 (50 mM Tris-HCL pH 7.4, 1 M NaCl, 1 mM EDTA, 1% NP-40, 0.1% SDS, 0.5% Na deoxycholate plus PI) and 1 time with wash buffer 2 (20 mM Tris−HCL pH 7.4, 10 mM MgCl$_2$, 0.2% Tween-20 plus PI). The beads were resuspended in wash buffer 2 and transferred to a new 1.5-mL Eppendorf tube. Samples were placed on a magnetic stand to remove the wash buffer. DNA was eluted and de-crosslinked in 1X TE plus 1% SDS, proteinase K, and 400 mM NaCl at 65 °C for 4 h. DNA was precipitated by phenol, chloroform, and isopropyl alcohol. Libraries were constructed by NEBNext Ultra II NEB Library Prep Kit and NEB-Next Multiplex oligos for Illumina. Publicly available ChIP-seq data were downloaded and processed following ENCODE guidelines. The analysis code and detailed description can be found in Yang, et.al.[59,60]. Briefly, reads were mapped to the human genome hg19(GRCh37-lite) by BWA (version 0.7.12-r1039, default parameter). Duplicated reads were marked with Picard (version 2.6.0-SNAPSHOT), and only non-duplicated reads were kept by samtools (parameter "-q 1 -F 1024" version 1.2). MACS2 (version 2.1.1.20160309) was used for peak calling. To ensure replicability, reproducible peaks for each group were finalized as only peaks retained if called with a stringent cutoff (-q 0.05) in one sample and at least called with a lower cutoff (-q 0.5) in the other samples. Homer was used for motif analysis[61]. HOXA9-bound peak files were provided in Supplementary Data 4. Raw ChIP-seq data were deposited in the dataset GSE215928 in GEO.

## ATAC-seq

Briefly, 10,000 SEM cells were treated with a transposition mix consisting of 22 μL nuclease-free water; 25 μL 2 × TD Buffer; 2.5 μL TDE1 transposase; and 0.5 μL of 1% digitonin (added immediately before removing the supernatant from cell pellets), followed by incubation at 37 °C on an Eppendorf ThmorMixer at 300 rpm for 30 minutes. DNA was purified using the MinElute Reaction Cleanup Kit (Qiagen, 28204) and amplified for 5 cycles using the PCR mix composed of 10 μL transposed DNA, 2.2 μL nuclease-free water, 6.25 μL 10 μM barcoded Nextera primer-F, 6.25 μL 10 μM barcoded Nextera primer-R, 0.3 μL 100 × SYBR Green I (Invitrogen, S-7563), and 25 μL NEBNext® High-Fidelity 2 × PCR Master Mix (NEB, M0541). PCR products were subsequently re-amplified for another 6 PCR cycles, followed by purification using SPRIselect beads (Beckman Coulter, B23317). Paired-end sequencing (2 × 100 bp) was performed using the Illumina HISeq 4000 platform. Paired-end reads were analyzed using cutadapt (version 1.18)[62] for adaptor trimming and then mapped to the human hg19 genome by Bowtie2 (version 2.2.9)[63]. Peak calling was performed on each sample by using MACS2[64] with default parameters, and peaks were then merged by BEDtools (version 2.25.0)[65] to retain non-overlapped regions for identifying differentially enriched peaks, using ABSSeq under the aFold model[66] with read count from HTSeq[67]. A cutoff of the adjusted P-value (FDR) < 0.05 and |$log_2$ fold change| ≥1 was used to define high-confidence ATAC-seq peaks. Enriched regions were mapped to the nearest gene in human hg19 by Homer[61]. ATAC-seq peak files were provided in Supplementary Data 5.

## Flow cytometry

To determine the percentage of cells targeted with a sgRNA coupled with CFP, cells grown in suspension culture were collected and filtered through a 70-micron cell strainer before sorting with flow cytometry. DAPI was added to the cell suspension to exclude dead cells. Fluorescence from CFP was detected using a blue-violet laser at a wavelength of 445 nm.

## Mouse studies

NOD.Cg*Prkdc*scid*Il 2rg*tm1Wjl/SzJ (NSG) mice were purchased from the Animal Resource Center at St. Jude Children's Research Hospital. NSG male mice are showing aggressive behavior than female settings particularly implanted with leukemia and developed disease progression. To avoid potential physical fighting and injury followed with infection, only female mice were included in this type of studies as a default setting. All mice were maintained in sterilized conditions with a temperature between 20 °C and 23 °C and humidity between 40% and 60%. Protocols for mouse studies were approved by the St. Jude Children's Research Hospital Institutional Animal Care and Use Committee. Aliquots of $2 \times 10^6$ SEM cells, comprising $1.6 \times 10^6$ SEM control cells and $0.4 \times 10^6$ SEM-sgNT CFP cells, SEM-dCas9-KRAB-sgFLT3-DE-1 CFP cells or SEM-Cas9-sgFLT3-DE-1 cells were injected into NSG female mice

(aged 8-12 weeks old) via tail vein injection. Starting two weeks after injection, the percentage of CFP-positive SEM cells in the peripheral blood was monitored using flow cytometry (cells were stained with antibodies to human CD45-FITC [BD Pharmingen #555482], human CD19-PE [BD Pharmingen #340364], mouse CD45-APC-Cy7 [BD Pharmingen #557659], and mouse Ter119-PerCP-Cy5.5 [BD Pharmingen #560512]). The mice were euthanized by $CO_2$ treatment and cervical dislocation following the approved animal protocol, when the mice reached a threshold of engraftment (generally above 80% human leukemia cells in the peripheral blood). Most mice show no overt symptoms at this point, which we consider a humane endpoint of the study. We also euthanized mice immediately if they had any of the following severe clinical signs: dragging hind limbs, inability to eat or drink, failure or delay to the right when placed on the back, dyspnea, bleeding from GI or respiratory tract, or any neurological signs such as circling head tilt, or seizures.

## Statistical analysis

Statistical analyses were performed using GraphPad Prism 9.0. For Q-PCR experiments, $p$ values were calculated using a two-tailed un-paired $t$-test from 3 independent biological replicates. For animal experiment, three or four female NSG mice were used for transplantation of control and CRISPR targeted cells. The $p$ values were calculated using a two-tailed un-paired $t$-test.

## Reporting summary

Further information on research design is available in the Nature Portfolio Reporting Summary linked to this article.

## Data availability

The raw CRISPR screen, Capture-C, ChIP-seq, ATAC-seq, and RNA-seq data generated in this study have been deposited in NCBI GEO under super series (GSE215928). The HiC publicly available data used in this study are available in the GEO (GSE138862)[25]. ATAC-seq publicly available data used in this study are available in the GEO (GSE74912);[34] (GSE129066);[68] (GSE153237)[69]. Transcription factor ChIP-seq publicly available data used in this study are available in the GEO (GSE117864)[70]. H3K27ac ChIP-seq publicly available data used in this study are available in the GEO (GSE17312)[71], (GSE80779)[35], (GSE65138)[36], (GSE79899)[72], (GSE109492)[73], (GSE137652);[74] (GSE111293)[75]. (GSE111179)[76]. The remaining data are available within the Article, Supplementary Information. Source data are provided with this paper.

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

## Competing interests

The authors declare no competing interests.
