## [Peer Review File · Nature Communications]

Systematic characterization of the HOXA9 downstream targets in MLL-r leukemia by noncoding CRISPR screensReviewers' Comments:

Reviewer #1:

Remarks to the Author:

Summary: Wright et al. present an investigation of HOXA9 downstream regulatory mechanisms in leukemic cells using a dropout CRISPR screen. The authors propose that FLT3 and other genes crucial for leukemic cell survival are direct targets of HOXA9. FLT3 as a downstream target of HOXA9 has been previously shown (Gwin et al. J Immunol 2010; Huang et al. Blood 2012). However, the study has its merits for discovery of new potential targets of HOXA9, such as XBP1.

Comments:

1. Considering the current state of the Discussion section, the study has limited clinical impact. FLT3, the only current downstream target of HOXA9 that is targetable in leukemia, has already been identified as a target of HOXA9 in prior studies, as mentioned above. In leukemia patients with overexpression of HOXA9, do the authors suggest that the newly-identified targets such as XBP1 can be targeted? Are there any inhibitors of newly-identified targets being developed in pre-clinical studies and/or clinical trials?
2. While the data on the interaction of FLT3 and HOXA9 is strong and convincing, the data on the newly-identified targets such as XBP1 is much more limited. Similar experiments conducted for FLT3 should also be conducted for these newly-identified targets. For example, is the expression of newly-identified targets dependent on looping factor CTCF? Is chromatin accessibility of these newly-identified targets different in MLL-r ALL cells compared with non-MLL-r cells? A complete investigation of these newly-identified targets will strengthen the study and fulfil the aim of the study mentioned in the title of the paper: "Comprehensive and functional interrogation of the HOXA9 downstream regulation mechanism in MLL-r leukemia".
3. In some experiments, only one MLL-r cell line was used. Eg. "As expected, MLL-r SEM cells exhibited superior sensitivity to pharmacological inhibition of FLT3 by Gilteritinib (LC50: 54 nM), when compared to Nalm6 (LC50: 1,852 nM), REH (LC50: 1,523 nM), and 697 (LC50: 1,729) cells (Figure 4I). Supporting experiments with multiple MLL-r cell lines such as RS4;11 could strengthen the data.
4. Title of the paper: "Comprehensive and functional interrogation of the HOXA9 downstream regulation mechanism in MLL-r leukemia". This is not compatible with the future directions mentioned above: "extensive work is also warranted to define the detailed regulation target genes of other top enriched sgRNAs (e.g., sg8105 and sg6931, as well as others)." A comprehensive and functional interrogation of the HOXA9 downstream regulation mechanism should also include data on the other top enriched sgRNAs (sg8105, sg6931, etc).
5. P-values or significance level should always accompany results whenever a difference is described. Eg. Figure 3B does not indicate any significance level and the results section states: "Cas9-mediated genome editing by these two sgRNAs resulted in significant downregulation of expression of FLT3 expression and the adjacent PAN3, as compared with that in non-target control (Figure 3B)."
6. "Consistent with previous reports, FLT3 expression was significantly higher in MLL-r leukemia cell lines (n=11) than in non-MLL-r leukemia cell lines (n=89) (p<0.0001)." What is the expression of HOXA9 in MLL-r leukemia vs non-MLL-r leukemia? The authors should provide data indicating that MLL-r cell lines have higher HOXA9 and higher FLT3.
7. "Of note, according to the TCGA database, high FLT3 expression was associated with poor overall survival in a cohort of 106 patients with AML (Supplementary Figure S4A-S4D)." Was high expression of HOXA9 associated with poor survival in these patients? Did the other newly-identified targets of HOXA9 (eg. XBP1) also show association with poor survival? Was the expression of HOXA9 and FLT3 correlated? Can the authors use another independent and publically available datasets to verify these results?
8. Abstract: "The HOXA9 protein is a poor therapeutic target as it lacks targetable pocket domains." Firstly, this was not substantiated with any relevant references anywhere in the main text. Secondly, as the authors point out in the Introduction, MEIS1 is a cofactor for HOXA9 and "co-expression of HOXA9 and MEIS1 is sufficient to induce leukemogenesis". Is there any evidence showing that MEIS1

is not a suitable therapeutic target? If HOXA9 is an important protein in leukemogenesis, then targeting MEIS1 can produce similar effects as directly targeting HOXA9 and can be a better strategy than targeting newly-discovered HOXA9 targets.

9. Introduction: "The oncogenic activity of HOXA9 was first suggested by the clinical correlation between high HOXA9 expression and poor outcomes." The authors should provide references here.

10. Results: "The FLT3 gene is well known to be essential for survival in MLL-r subtypes based on genomic and genetic evidence." The authors should provide references here.

11. Results: "It has been reported that double-stranded breaks may induce cell death and may cause the HOXA9-binding motif-independent phenotype." The authors should provide references here.

12. Results: "It will be interesting to investigate the noncoding regulation of the gene by HOXA9 in a follow-up study in the future. Moreover, extensive work is also warranted to define the detailed regulation target genes of other top enriched sgRNAs (e.g., sg8105 and sg6931, as well as others)." The future directions should not be described in the Results section. This should be described in the Discussion Section.

13. The authors should carefully check their paper for typos. Eg. Results: "Also, MLL-r leukemia cell lines (n=8) were more dependency on FLT3".

14. The prior reports on HOXA9 regulation of FLT3, as mentioned above (Gwin et al. J Immunol 2010; Huang et al. Blood 2012) should be included as references and discussed as appropriate.

Reviewer #2:

Remarks to the Author:

In this paper, the authors use a CRISPR dropout screen to identify functionally relevant HOXA9 binding sites that are important for the survival of the MLL-rearranged acute lymphoblastic leukemia SEM cell line. They identify a HOXA9 binding site upstream of FLT3, and demonstrate its importance in FLT3 expression and cell growth. The authors also characterise several other HOXA9 binding sites to identify the most likely target genes being regulated. Overall this is an interesting screen, which has the potential to dissect the key targets of HOXA9 in ALL cells. However, for this reviewer, the data is underexplored, as it would have been nice to see more of the HOXA9 peaks investigated.

Major points:

1. The authors should acknowledge other work in the field that has been done already to functionally explore the role of HOXA9 at enhancers, in particular Sun et al 2018 (<https://doi.org/10.1016/j.ccell.2018.08.018>) and Zhong et al 2018 (<https://doi.org/10.1182/bloodadvances.2018025866>).

The authors cite Milne et al 2002 and Milne et al 2005 as evidence that MLL-fusion proteins assemble a complex of proteins at the HOXA9 locus. While these papers nicely show that wild type and MLL-fusion proteins bind at HOXA9, they don't explore the binding of any co-factors – there are plenty of other papers which show the complexes assembled by MLL-FPs which the authors could cite.

2. Considering the authors targeted 1806 peaks with their sgRNA dropout screen, and observed dropout of 522 of these, it is surprising that they only selected six peaks for further study. Therefore, I'm not sure it's fair to describe this as a comprehensive interrogation of HOXA9 downstream regulation – given the number of experiments devoted to FLT3 regulation, I would think it would be better to reduce the scope of the paper, and note FLT3 as the focus of the paper in the title and abstract. Alternatively, more work should be done on exploring other hits from the screen. For example, more mechanistic studies of the regulation of XBP1, JUN, BACHH1 (e.g. Capture-C to demonstrate the interaction between the HOXA9 peaks and gene promoters) would strengthen the functional relevance of these HOXA9 loci.

3. Experimental design: In the methods (page 16 lines 345-346) the authors say that they targeted

1806 HOXA9 peaks that carried consensus motifs, but they don't say how they filtered the HOXA9 ChIP-seq peaks to remove any peaks that don't carry consensus motifs – it's unlikely that all HOXA9 peaks would have the consensus, but if so, the authors still need to confirm that they had checked this.

Page 6 lines 120-121: "As the HOXA9-bound peaks were located at distal noncoding regions". This is the first time this has been mentioned – where is the data to support the assertion that all peaks are at distal noncoding regions? The authors previously said that most of the 1806 peaks targeted were present at cis-regulatory elements, with ~20% at promoters (Supplementary Figure S1A), so what is the basis for saying that the peaks are at distal non-coding regions? Additionally, of these distal peaks, what proportion are found at enhancers? This would give further support to the idea they are functionally relevant for gene activation.

Page 6 lines 123-126 (Supplementary Figure S2C); page 11 lines 227-232: The authors pick several sgRNA to validate by CPA – but only give information about the target loci for some of them (e.g. no information on sg6931). Where are these HOXA9 peaks (are they at promoters or distal loci)? Do they look like enhancers? These are referred to as being HOXA9-bound noncoding regions, but there's no evidence provided to support this assertion. It would be nice to see a couple of (zoomed in) examples of where the sgRNA binds relative to the HOXA9 peak (and consensus motif), for example for the sgRNAs targeting the FLT3 enhancer.

Page 8 lines 167-169: The big piece of data that's essential for arguing that h-FLT3 is important for upregulating FLT3 expression is to mutate the locus and assess its effect on FLT3 expression – but you don't do this until Figure 3, after you've already concluded the locus is important! This seems like an odd order in which to present this.

4. Considering how much the conclusions of the paper are dependent on the sgRNA-targeted mutation of HOXA9 binding sites, it's surprising that there is little or no validation that HOXA9 binding is lost in the transduced cells. I recognise that ChIP for HOXA9 is extremely difficult, but proxy assays could be used – for example, ChIP showing a reduction in H3K27ac, or ATAC-seq, at targeted enhancer loci (such as h-FLT3). If HOXA9 binding is functionally important, its loss should have effects on local chromatin.

As highlighted by the authors in several places, these HOXA9-bound peaks are also bound by a number of other transcription factors. Therefore, the CRISPR dropout screen can't be used to directly ascribe a functional role to HOXA9 binding at target loci. The authors can say that these DNA elements are important to growth/survival, but they can't say that HOXA9 binding specifically is important, as it may be one or more of the other TFs also seen to bind at them.

5. There are several instances of the same data being shown twice in the paper (or virtually the same experiment being conducted twice, with equivalent data being shown). This is particularly prevalent for the growth competition assays. For example, Figure 3C: how is this experiment different from Supplementary Figure S2C? These are both CPA time-courses in SEM cells targeting h-FLT3. Figure 4H appears to be the same data as Supplementary Figure S4A. Figure 6H: This is the same as Supplementary Figure S2C (for SEM cells). The authors also target the same loci in MOLM13 cells, but don't explain the relevance of this, or comment on the result. There is an important point about these effects not being specific to SEM cells, but the authors need to make it.

Figure 5B/C: There is no need to repeat the same data in these figures as a pool as well as separately – if the authors really want to do this, they should move one set of the bar charts into supplementary.

6. Figure 2C/D: It is very hard to interpret this Hi-C/HiChIP data – please annotate to make it clearer what you are trying to show (i.e. the interaction between the FLT3 promoter and enhancer). The resolution of these techniques is not ideal for exploring how CTCF degradation affects the interaction

between the HOXA9 binding site and FLT3 promoter.

It would be more informative to look at the effect of CTCF degradation by Capture-C. Capture-C shows that the region interacting with the promoter is much broader than just the CTCF peak (without annotation it's hard to see where the CTCF peak lies). Does the entirety of this interaction disappear when upon CTCF degradation, or just the interaction at the CTCF peak? This may help to explain why CTCF degradation does not affect FLT3 expression. This would also complement the Capture-C experiment with the sgFLT3-DE1/2 dCas9, by demonstrating whether all or only some of the interactions are dependent on CTCF.

7. Figure 6E: The authors need to make it clear that this is pathway analysis of the overlap DEGs that are affected both by Dox and 5-Ph-IAA treatment. I'm not sure what the logic is for only looking at genes that are sensitive to both HOXA9 overexpression and loss of that overexpression? This seems like a very complicated experiment to do rather than just knocking down HOXA9 (which would more directly test a requirement for HOXA9). Did the authors try this?

Figure 6G: Overexpression of HOXA9 results in strong downregulation of XBP1, but sg3159 (which presumably disrupts HOXA9 binding) also downregulates XBP1 expression. On the face of it, these results are counterintuitive – how do the authors explain this? This argues that the role of HOXA9 is more complex than presented, and the authors should discuss possible interpretations of the data.

8. Figure 3H: OCI-AML2 cells appear to be less sensitive to sgRPS19 as well as sgFLT3-DE-1/2, suggesting that these cells may be more resistant to editing in general. If the authors wish to conclude that these cells are less dependent on the h-FLT3 enhancer, they should sequence the targeted locus in the transduced cells to demonstrate that the region has been successfully mutated.

Supplementary Figure S6C: sgJUN1/2 only very weakly phenocopy sg2571 in the CPA. This suggests that this either the sgRNA sequences targeting JUN are ineffective (which can be assessed by gDNA sequencing and/or western blotting for JUN) or that the sg2571 locus has additional/alternative regulatory functions. Please discuss.

Page 11 lines 249-250: "among the nearby genes of another candidate sgRNA (sg3243), only expression of the BAHCC1 gene was decreased by Cas9-sg3243". Where are the data to support this statement? I'm not sure why the authors show the different TFs bound at the BAHCC1 locus (Figure 7A) as they don't make any reference to this in the text.

9. Figure 8 is entirely speculative and barely discussed at all in the text. The suggestion in the figure that deletion of a HOXA9 binding site would produce a different phenotype to HOXA9 knock-out is an interesting one – but nowhere is this discussed. Of course, an alternative explanation (not addressed in the paper) is that mutation of the sequence disrupts the binding of multiple TFs (not just HOXA9), and perhaps one or more of these TFs has a (more?) important role in gene activation.

Minor points:

There are a lot of typographical and grammatical errors in this paper which need to be addressed. In several cases the authors refer to several figure panels in one go, where it would be much easier for the reader to follow if individual panels were used to make specific points.

1. Supplementary Figure S1D: what is the difference between the motif enrichment on the left and the right? Are these called using different algorithms? There's no detail in the legend. Why does this figure refer to 3774 HOXA9 peaks and 2820 MEIS1 peaks, when the text refers to 1806 HOXA9 and 6725 MEIS1 peaks?

2. Page 6 line 112: "a known distal enhancer of the FLT3 locus" needs a citation.

3. Page 6 line 115 (Supplementary Fig S2A): unless I have misunderstood (in which case you need to

explain this more clearly) this is not motif analysis. Please rephrase. What is your logic for looking at these transcription factors in particular?

4. Page 7 lines 138-139: needs proof-reading.

5. Page 7 line 145: you need to add "in SEM cells (Figure 2B)." You are using data from multiple MLLr cell lines here interchangeably – I think this is justified, but you need to make sure you are explicit about which dataset you are referring to at each point.

6. Page 8 lines 160-169: you should refer to individual panels of Supplementary Figure S4 to support specific statements – not just cite them all together at the end. This makes it much harder for the reader to assess the data.

7. Page 8 line 162: "dependency" should be "dependent".

8. Page 11 line 239: "deferentially" should be "differentially".

9. Page 11 lines 246-247: "We could match the sg3159 editing effect with transcriptionally affected genes located in the same topological associated domain (Figure 6G)." I think there must be a mistake in this sentence, because it doesn't match the conclusion from Figure 6G.

10. Page 13 lines 297-298: needs proof-reading.

11. Page 16 line 355: "illumine" should read "Illumina".

12. Page 26 lines 663-664: needs proof-reading.

13. Pages 26-27: Legend is missing for Fig2D.

14. Figure 6B: The labels for the bands on the western need to be more clearly aligned (> symbols are not ideal) – for the lower blot it's unclear which band the arrow is pointing at.

15. Supplementary Figure S3A: legend should read "Depletion of CTCF specifically decreased the chromatin accessibility at the CTCF binding site of the h-FLT3 enhancer". Use of the term "target site" is ambiguous (do you mean the HOXA9 binding site, or sgRNA target site?).

16. Supplementary Figure S6A: This is referred to in the text in the context of sg2571/JUN, but the figure appears to be about sg8105.

Reviewer #3:

Remarks to the Author:

In the manuscript entitled "Comprehensive and functional interrogation of the HOXA9 downstream regulation mechanism in MLL-r leukemia", Wright A & Zhao X et al. provided relevant data on the downstream functional effectors of HOXA9 which could represent potential therapeutic targets for MLL-r leukemias.

Major comments:

- Authors should explain which cell lines they reanalyzed using CHIP assays. In the results, they only named B-ALL SEM and "other cell lines".
- Could the authors further explain and validate the differences among common binding peaks of HOXA9 and MEIS1, as well as those that are specific for HOXA9?
- How did you define the subgroups of AML patients according to FLT3 expression (Figure S4D)?
- The authors should explain with more details the approach to OE and KD genes (Figure 6A).
- BAHCC1 is a potential target that should be validated in an in vivo model.
- The authors used a cohort of 106 AML patients from TCGA. However, these results should be validated in other cohorts.

REVIEWER COMMENTS

Reviewer #1, expertise in epigenomics, multi-omics, haematological malignancies (Remarks to the Author):

Summary: Wright et al. present an investigation of HOXA9 downstream regulatory mechanisms in leukemic cells using a dropout CRISPR screen. The authors propose that FLT3 and other genes crucial for leukemic cell survival are direct targets of HOXA9. FLT3 as a downstream target of HOXA9 has been previously shown (Gwin et al. J Immunol 2010; Huang et al. Blood 2012). However, the study has its merits for discovery of new potential targets of HOXA9, such as XBP1.

Response: Thank you for summarizing our key findings and recognizing the value of this study.

Comments:

1. Considering the current state of the Discussion section, the study has limited clinical impact. FLT3, the only current downstream target of HOXA9 that is targetable in leukemia, has already been identified as a target of HOXA9 in prior studies, as mentioned above. In leukemia patients with overexpression of HOXA9, do the authors suggest that the newly-identified targets such as XBP1 can be targeted? Are there any inhibitors of newly-identified targets being developed in pre-clinical studies and/or clinical trials?

Response: We thank the reviewer for this comment. We agree with the reviewer that revealing potential clinical applications is very important. We realize that none of the current candidates are associated with FDA-approved drugs in the clinic. During the revision, we aim to identify more targets. To this end, we have established the inducible TetOn system to express HOXA9-HA cDNA upon doxycycline treatment in MLL-r leukemia cells. The no-drug treatment groups didn't show any leaking and served as a perfect isogenic negative control. By utilizing this new model system, we have successfully conducted a reproducible ChIP-seq and identified 229 high-quality and reproducible HOXA9-binding peaks, including the most significant peak at the distal enhancer of FLT3 (FLT3 DE1), which was identified by us in the first submission. Given the peak number variation was observed between our ChIP-seq and the publicly available dataset (229 vs. 1,806), we generated a new noncoding CRISPR library against all 229 HOXA9-bound peaks and then conducted a new dropout CRISPR screen. In addition to FLT3, we identified a novel functional HOXA9-binding peak in the intronic region of CDK6, which is a plausible intronic enhancer looped to the CDK6 promoter in MLL-r leukemia cells. Targeting this peak by CRISPR/Cas9 or dCas9-KRAB (CRISPRi) significantly reduced chromatin accessibility and leukemia cell fitness. These new data were shown as **New Figure 6** (below).

These efforts suggest that targeting CDK6 by FDA-approved CDK4/6 inhibitors could potentially benefit the MLL-r leukemia therapy alone or synergy with other drugs. Although the current scope of our study is not directly focusing on preclinic and clinical treatment, we are very interested in testing these hypotheses in future research.

New Figure 6. Functional interrogation of novel HOXA9-bound targets by genome editing. (A) Competitive proliferation assay was conducted in Cas9-expressing SEM cells targeted with sgRNAs against HOXA9-bound sites close to the genes as PEBP4, ZCCNC7, AHI1, RUNX1, DCAF11, CDK6, NDUFS8, and MAN1C1. The sgRPS19 and sgFLT3-DE-1 sgRNA served as positive controls. Disruption of the loci targeted by these novel enriched sgRNAs led to retarded cell growth of SEM cells in a time-dependent manner. The percentage of cell numbers was normalized to CFP+ control cells infected with non-target sgRNA (sgNT). (B) Competitive proliferation assay was conducted in Cas9-expressing MOLM13 cells targeted with sgRNAs against HOXA9-bound sites close to the genes as PEBP4, ZCCNC7, AHI1, RUNX1, DCAF11, CDK6, NDUFS8 and MAN1C1. The sgRPS19 and sgFLT3-DE-1 sgRNA served as positive controls. Disruption of the loci targeted by these novel enriched sgRNAs led to retarded cell growth of MOLM13 cells in a time-dependent manner. The percentage of cell numbers was normalized to CFP+ control cells infected with non-target sgRNA (sgNT). (C) Competitive proliferation assay was conducted in dCas9-KRAB-expressing SEM cells targeted with sgRNAs against HOXA9-bound sites close to the genes as RUNX1, DCAF11, and CDK6. The sgFLT3-DE-1 sgRNA served as positive control. (D) Competitive proliferation assay was conducted in dCas9-KRAB-expressing MOLM13 cells targeted with sgRNAs against HOXA9-bound sites close to the genes as RUNX1, DCAF11, and CDK6. The sgFLT3-DE-1 sgRNA served as positive control. (E) Characterization of the chromatin conformation change upon dCas9-KRAB targeting against the HOXA9-bound site in the intron of CDK6. HiC, HOXA9 ChIP-seq, H3K27ac ChIP-seq, and BRD4 ChIP-seq tracks were shown to characterize the epigenetic status of the HOXA9-bound site. (F) Q-PCR was conducted to quantify the transcription decrease of CDK6 when CRISPRi targeted the HOXA9-bound site in the intron of CDK6 in SEM cells. (G) Q-PCR was conducted to quantify the transcription decrease of CDK6 when CRISPRi targeted the HOXA9-bound site in the intron of CDK6 in MOLM13 cells. (H) Total RNA-seq was carried out using SEM cells targeted with sgCDK6 against the HOXA9-bound site in the CDK6 intron. Differential gene expression was defined by $FDR < 0.01$. The CDK6 expression is the top hit. Data are shown as mean values \pm SEM of three biological replicates. **: $p < 0.01$, ****: $p < 0.0001$. P values were estimated using a two-tail t-test.

2. While the data on the interaction of FLT3 and HOXA9 is strong and convincing, the data on the newly-identified targets such as XBP1 is much more limited. Similar experiments conducted for FLT3 should also be conducted for these newly-identified targets. For example, is the expression of newly-identified targets dependent on looping factor CTCF? Is chromatin accessibility of these newly-identified targets different in MLL-r ALL cells compared with non-MLL-r cells? A complete investigation of these newly-identified targets will strengthen the study and fulfil the aim of the study mentioned in the title of the paper: “Comprehensive and functional interrogation of the HOXA9 downstream regulation mechanism in MLL-r leukemia”.

Response: We thank the reviewer for this comment. To address the reviewer’s questions, we now provide the following data.

1). In this revision, we have conducted a successful HOXA9 ChIP-seq in MLL-r SEM cells and then performed a new dropout CRISPR screen against the 229 HOXA9-bound peaks. Therefore, in addition to FLT3, we systematically validated another candidate locus in the intronic region of CDK6 by competitive proliferation assay, chromatin conformation characterization and chromatin accessibility profiling analysis (shown above, **New Figure 6**).

2). To examine the chromatin accessibility of the newly identified targets in MLL-r and non-MLL-r cells, we utilized the publicly available CHIP-seq data. We focused on the two major targets we reported in this study, FLT3 and CDK6. We did not see a significant difference in ATAC-seq activity between MLL-r and non-MLL-r cells (shown below).

Figure 2A. ATAC-seq profiling of FLT3 locus in MLL-r and non-MLL-r leukemia cell lines.

Figure for revision-1. ATAC-seq profiling of CDK6 locus in MLL-r and non-MLL-r leukemia cell lines.

3). To address the question of whether the expression of newly identified targets is dependent on looping factor CTCF, we take advantage of our newly established acute protein degradation model of CTCF in SEM cells (Genome Biol. 2023 Jan 26;24(1):14. doi: 10.1186/s13059-022-02843-3). Upon efficient and acute degradation of CTCF protein, we conducted Q-PCR and immunoblotting assays, suggesting no changes. In addition, we performed an ATAC-seq assay to characterize chromatin

accessibility change and Capture-C to monitor chromatin conformation change. Our results suggest loss of CTCF does not impact chromatin accessibility nor chromatin looping at the target locus. Moreover, we further evaluated the transcription change of other validated candidates compared to RBM45, the known downstream target of CTCF. All of the HOXA9-bound targets indicate minor expression changes upon CTCF loss. In summary, our efforts and complete investigation of these newly identified targets will strengthen the study and fulfil the aim of the study. These data were shown below (**New Figure 2, and Figure for revision-2**).

New Figure 2. (A) schematic diagram of auxin-inducible degron (AID) system to target and acutely deplete CTCF protein. (B) Immunoblotting was conducted to confirm the acute protein degradation of CTCF in the presence of auxin for 24 hours. Immunoblotting was also performed to detect the expression of FLT3 protein upon acute degradation of CTCF. GAPDH was included as an internal control. (C) Q-PCR was used to examine the mRNA expression of FLT3 upon CTCF protein degradation. Data are shown as mean values \pm SEM of three biological replicates (center of the error bar). P values were estimated using a two-tail t-test. (D) Capture-C and ATAC-seq were conducted to characterize the chromatin accessibility change and chromatin conformation change, respectively. The biotin-labeled DNA oligos against the FLT3 promoter were used as baits to quantify chromatin looping between the FLT3 promoter and the h-FLT3 enhancer.

Figure for revision 2. A. Total RNA-seq analysis to characterize HOXA9’s target gene expression upon acute CTCF loss. RBM45 is a positive control for a known CTCF’s target gene. Data are shown as mean values \pm SEM of four biological replicates. **: $p < 0.01$, ***: $p < 0.001$, ****: $p < 0.0001$. P values were estimated using a two-tail t-test.

3. In some experiments, only one MLL-r cell line was used. Eg. “As expected, MLL-r SEM cells exhibited superior sensitivity to pharmacological inhibition of FLT3 by Gilteritinib (LC50: 54 nM), when compared to Nalm6 (LC50: 1,852 nM), REH (LC50: 1,523 nM), and 697 (LC50: 1,729) cells (Figure 4I). Supporting experiments with multiple MLL-r cell lines such as RS4;11 could strengthen the data.

Response: We thank the reviewer for this comment. We conducted an MTT assay and calculated the IC50 of RS411 as 841nM, which is between SEM and non-MLL-r ALL cell lines.

Figure for revision-3. Drug response test of JQ1 in MLL-r ALL RS411 cell line.

4. Title of the paper: “Comprehensive and functional interrogation of the HOXA9 downstream regulation mechanism in MLL-r leukemia”. This is not compatible with the future directions mentioned above: “extensive work is also warranted to define the detailed regulation target genes of other top enriched sgRNAs (e.g., sg8105 and sg6931, as well as others).” A comprehensive and functional interrogation of the HOXA9 downstream regulation mechanism should also include data on the other top enriched sgRNAs (sg8105, sg6931, etc).

Response: We thank the reviewer for this comment. We agree with the reviewer that the previous title is not reflecting the entire content well. Therefore, we decide to change the title to “Systematic characterization of the HOXA9 downstream targets in MLL-r leukemia by noncoding CRISPR screens”.

5. P-values or significance level should always accompany results whenever a difference is described. Eg. Figure 3B does not indicate any significance level and the results section states: “Cas9-mediated genome editing by these two sgRNAs resulted in significant downregulation of expression of FLT3 expression and the adjacent PAN3, as compared with that in non-target control (Figure 3B).”

Response: We thank the reviewer for this comment. We now provide the *t*-test p values.

New Figure 3D. Q-PCR detection of gene expression demonstrated notable downregulation of FLT3 and PAN3 expressions in SEM cells upon CRISPR targeting. Data are shown as mean values \pm SEM of three biological replicates. P values were estimated using a two-tail t-test.

6. “Consistent with previous reports, FLT3 expression was significantly higher in MLL-r leukemia cell lines (n=11) than in non-MLL-r leukemia cell lines (n=89) ($p < 0.0001$).” What is the expression of HOXA9 in MLL-r leukemia vs non-MLL-r leukemia? The authors should provide data indicating that MLL-r cell lines have higher HOXA9 and higher FLT3.

Response: We thank the reviewer for this comment. We now provide the Depmap RNA-seq data indicating that HOXA9 expression is higher in MLL-r cells compared to non-MLL-r settings.

Figure for revision-4. Summary of HOXA9 expression in MLL-r and non-MLL-r cell lines using Depmap expression datasets. P values were estimated using two-tail *t*-test.

7. “Of note, according to the TCGA database, high FLT3 expression was associated with poor overall survival in a cohort of 106 patients with AML (Supplementary Figure S4A-S4D).” Was high expression of HOXA9 associated with poor survival in these patients? Did the other newly-identified targets of HOXA9 (eg. XBP1) also show association with poor survival? Was the expression of HOXA9 and FLT3 correlated? Can the authors use another independent and publically available datasets to verify these results?

Response: We thank the reviewer for this comment. We now provide the survival data from the TCGA AML cohort, which indicates that the high expression of HOXA9 is associated with poor survival (A), similar to the observation for FLT3. TARGET and BEAT cohorts (B and C) also observed a similar trend. We did not detect significant correlations from other validated candidates. These data were shown as **New Supplementary Figure S2**.

Also, we further analyzed the expression correlation of HOXA9 and FLT3 in the TCGA AML cohort (P-value=2e-05 and R=0.32). The figure below shows that the correlation will be even higher if we only examine the HOXA9-expressing patients.

Figure for revision-5. Expression correlation between HOXA9 and FLT3 in TCGA AML cohort (GEP1A2).

8. Abstract: “The HOXA9 protein is a poor therapeutic target as it lacks targetable pocket domains.” Firstly, this was not substantiated with any relevant references anywhere in the main text. Secondly, as the authors point out in the Introduction, MEIS1 is a cofactor for HOXA9 and “co-expression of HOXA9 and MEIS1 is sufficient to induce leukemogenesis”. Is there any evidence showing that MEIS1 is not a suitable therapeutic target? If HOXA9 is an important protein in leukemogenesis, then targeting MEIS1 can produce similar effects as directly targeting HOXA9 and can be a better strategy than targeting newly-discovered HOXA9 targets.

Response: We thank the reviewer for this comment. We apologize for the missing reference. What we want to emphasize is that most transcription factors were, for a long time, considered undruggable targets because of the absence of binding pockets for direct targeting, including HOXA9 and MEIS1. We agree with the reviewer that targeting MEIS1/HOXA9 interaction axis is an alternative approach to cure MLL-r leukemia. However, we are not aware of any reported success targeting MEIS1.

9. Introduction: “The oncogenic activity of HOXA9 was first suggested by the clinical correlation between high HOXA9 expression and poor outcomes.” The authors should provide references here.

Response: We thank the reviewer for this comment. We apologize for the missing reference. We have updated the reference accordingly.

10. Results: “The FLT3 gene is well known to be essential for survival in MLL-r subtypes based on genomic and genetic evidence.” The authors should provide references here.

Response: We thank the reviewer for this comment. We apologize for the missing reference. We have updated the reference accordingly.

11. Results: “It has been reported that double-stranded breaks may induce cell death and may cause the HOXA9-binding motif-independent phenotype.” The authors should provide references here.

Response: We thank the reviewer for this comment. We apologize for the missing reference. We have updated the reference accordingly.

12. Results: “It will be interesting to investigate the noncoding regulation of the gene by HOXA9 in a follow-up study in the future. Moreover, extensive work is also warranted to define the detailed regulation target genes of other top enriched sgRNAs (e.g., sg8105 and sg6931, as well as others).” The future directions should not be described in the Results section. This should be described in the Discussion Section.

Response: We thank the reviewer for this comment. We agree with the reviewer that this part should be moved to the discussion section.

13. The authors should carefully check their paper for typos. Eg. Results: “Also, MLL-r leukemia cell lines (n=8) were more dependency on FLT3”.

Response: We apologize for the typos. We have carefully revised the manuscript and corrected the typos.

14. The prior reports on HOXA9 regulation of FLT3, as mentioned above (Gwin et al. J Immunol 2010; Huang et al. Blood 2012) should be included as references and discussed as appropriate.

Response: We thank the reviewer for this comment. We apologize for the missing reference. We have updated the reference accordingly.

Reviewer #2, expertise in MLL and epigenetics (Remarks to the Author):

In this paper, the authors use a CRISPR dropout screen to identify functionally relevant HOXA9 binding sites that are important for the survival of the MLL-rearranged acute lymphoblastic leukemia SEM cell line. They identify a HOXA9 binding site upstream of FLT3, and demonstrate its importance in FLT3 expression and cell growth. The authors also characterise several other HOXA9 binding sites to identify the most likely target genes being regulated. Overall this is an interesting screen, which has the potential to dissect the key targets of HOXA9 in ALL cells. However, for this reviewer, the data is underexplored, as it would have been nice to see more of the HOXA9 peaks investigated.

Response: We thank the reviewer for this comment. In addition to FLT3, we also conducted experiments to validate the top 8 candidates further. We characterized the gene regulation of a new HOXA9-bound peak in the intron of CDK6 by competitive proliferation assay, chromatin conformation characterization and chromatin accessibility profiling analysis. These new data were shown as **New Figure 6**.

Major points:

1. The authors should acknowledge other work in the field that has been done already to functionally explore the role of HOXA9 at enhancers, in particular Sun et al 2018 (<https://doi.org/10.1016/j.ccell.2018.08.018>) and Zhong et al 2018 (<https://doi.org/10.1182/bloodadvances.2018025866>).

Response: We thank the reviewer for this comment. We apologize for the missing reference. We have updated the reference accordingly.

The authors cite Milne et al 2002 and Milne et al 2005 as evidence that MLL-fusion proteins assemble a complex of proteins at the HOXA9 locus. While these papers nicely show that wild type and MLL-fusion proteins bind at HOXA9, they don't explore the binding of any co-factors – there are plenty of other papers which show the complexes assembled by MLL-FPs which the authors could cite.

Response: We thank the reviewer for this comment. We apologize for the missing reference. We have updated the reference accordingly.

2. Considering the authors targeted 1806 peaks with their sgRNA dropout screen, and observed dropout of 522 of these, it is surprising that they only selected six peaks for further study. Therefore, I'm not sure it's fair to describe this as a comprehensive interrogation of HOXA9 downstream regulation – given the number of experiments devoted to FLT3 regulation, I would think it would be better to reduce the scope of the paper, and note FLT3 as the focus of the paper in the title and abstract. Alternatively, more work should be done on exploring other hits from the screen. For example, more mechanistic studies of the regulation of XBP1, JUN, BACHH1 (e.g. Capture-C to demonstrate the interaction between the HOXA9 peaks and gene promoters) would strengthen the functional relevance of these HOXA9 loci.

Response: We thank the reviewer for this comment. We agree to reduce the scope of the paper. Therefore, we want to focus on identifying HOXA9's targets by high-quality ChIP-seq and noncoding CRISPR screens, followed by a focus on two selected targets. To this end, we realize that our noncoding CRISPR screen against HOXA9's targets relies on an "unpublished" and GEO deposit of a HOXA9 ChIP-seq

dataset using MLL-r SEM cells. The quality of this ChIP-seq dataset is good based on motif analysis. However, a significant variation was also observed in two replicates. Therefore, to address this question, we have spent enormous efforts to set up the ChIP-seq assay using MLL-r SEM cells in our lab. In brief, to cope with the difficulty of ChIP-seq against endogenous HOXA9 by lacking good antibodies, we established the inducible TetOn system to express HOXA9-HA cDNA upon doxycycline treatment. The no-drug treatment groups show no leaking and serve as a perfect isogenic control. By utilizing this new model system, we have successfully conducted a reproducible ChIP-seq and identified 229 high-quality HOXA9-binding peaks, including the most significant peak at the distal enhancer of *FLT3* (FLT3 DE1), which was identified by us in the first submission. Given the peak number variation between our ChIP-seq and the publicly available one (229 vs. 1,806), we generated a new noncoding CRISPR library against all 229 HOXA9-binding peaks and then conducted a new dropout CRISPR screen. To our understanding, this is the best ChIP-seq data available so far. We systematically validated the top candidates in the following study by competitive proliferation assay, chromatin conformation characterization and chromatin accessibility profiling analysis. These efforts solidify our previous efforts to identify HOXA9's downstream targets in the MLL-r leukemia setting. These data were shown as **New Figure 6**.

New Figure 6. Functional interrogation of novel HOXA9-bound targets by genome editing. (A) Competitive proliferation assay was conducted in Cas9-expressing SEM cells targeted with sgRNAs against HOXA9-bound sites close to the genes as PEBP4, ZCCNC7, AHI1, RUNX1, DCAF11, CDK6, NDUFS8, and MAN1C1. The sgRPS19 and sgFLT3-DE-1 sgRNA served as positive controls. Disruption of the loci targeted by these novel enriched sgRNAs led to retarded cell growth of SEM cells in a time-dependent manner. The percentage of cell numbers was normalized to CFP+ control cells infected with non-target sgRNA (sgNT). (B) Competitive proliferation assay was conducted in Cas9-expressing MOLM13 cells targeted with sgRNAs against HOXA9-bound sites close to the genes as PEBP4, ZCCNC7, AHI1, RUNX1, DCAF11, CDK6, NDUFS8 and MAN1C1. The sgRPS19 and sgFLT3-DE-1 sgRNA served as positive controls. Disruption of the loci targeted by these novel enriched sgRNAs led to retarded cell growth of MOLM13 cells in a time-dependent manner. The percentage of cell numbers was normalized to CFP+ control cells infected with non-target sgRNA (sgNT). (C) Competitive proliferation assay was conducted in dCas9-KRAB-expressing SEM cells targeted with sgRNAs against HOXA9-bound sites close to the genes as RUNX1, DCAF11, and CDK6. The sgFLT3-DE-1 sgRNA served as positive control. (D) Competitive proliferation assay was conducted in dCas9-KRAB-expressing MOLM13 cells targeted with sgRNAs against HOXA9-bound sites close to the genes as RUNX1, DCAF11, and CDK6. The sgFLT3-DE-1 sgRNA served as positive control. (E) Characterization of the chromatin conformation change upon dCas9-KRAB targeting against the HOXA9-bound site in the intron of CDK6. HiC, HOXA9 ChIP-seq, H3K27ac ChIP-seq, and BRD4 ChIP-seq tracks were shown to characterize the epigenetic status of the HOXA9-bound site. (F) Q-PCR was conducted to quantify the transcription decrease of CDK6 when CRISPRi targeted the HOXA9-bound site in the intron of CDK6 in SEM cells. (G) Q-PCR was conducted to quantify the transcription decrease of CDK6 when CRISPRi targeted the HOXA9-bound site in the intron of CDK6 in MOLM13 cells. (H) Total RNA-seq was carried out using SEM cells targeted with sgCDK6 against the HOXA9-bound site in the CDK6 intron. Differential gene expression was defined by FDR<0.01. The CDK6 expression is the top hit. Data are shown as mean values \pm SEM of three biological replicates. **: p<0.01, ****: p<0.0001. P values were estimated using a two-tail t-test.

3. Experimental design: In the methods (page 16 lines 345-346) the authors say that they targeted 1806 HOXA9 peaks that carried consensus motifs, but they don't say how they filtered the HOXA9 ChIP-seq peaks to remove any peaks that don't carry consensus motifs – it's unlikely that all HOXA9 peaks would have the consensus, but if so, the authors still need to confirm that they had checked this.

Response: We thank the reviewer for this comment. In the new analysis, among the reproducible and high-quality HOXA9-bound ChIP-seq peaks, about 46.72% of targets contain the HOXA9 consensus motif compared with control peaks.

	HOXA9 (Homeobox)/HSC-Hoxa9-ChIP-seq (GSE33509)/Homer	P=1e-30	46.72% of targets with Motif
	RUNX2(Runt)/PCa-RUNX2-ChIP-Seq (GSE33889)/Homer	P=1e-29	54.15% of targets with Motif
	ETS1(ETS)/Jurkat-ETS1-ChIP-Seq (GSE17954)/Homer	P=1e-29	50.22% of targets with Motif

New Figure 1C. Motif analysis of HOXA9-bound peaks in SEM cells.

Page 6 lines 120-121: “As the HOXA9-bound peaks were located at distal noncoding regions”. This is the first time this has been mentioned – where is the data to support the assertion that all peaks are at distal noncoding regions? The authors previously said that most of the 1806 peaks targeted were present at cis-regulatory elements, with ~20% at promoters (Supplementary Figure S1A), so what is the basis for saying that the peaks are at distal non-coding regions? Additionally, of these distal peaks, what proportion are found at enhancers? This would give further support to the idea they are functionally relevant for gene activation.

Response: We thank the reviewer for this comment. We apologize for the incorrect description. The pie chart below shows that half of the HOXA9-bound peaks are resided in the intronic regions, 13.5% in promoters, and 29.1% in distal regions. We then overlap HOXA9-bound peaks with publicly available H3K27ac ChIP-seq in SEM (GSM3312817, 33128 Peaks). 73 (31.87%) out of 229 peaks were overlapping H3K27ac peaks. 28 (26.4%) of the distal HOXA9 peaks (106) overlapping H3K27ac peaks.

New Figure 1B. Genomic distribution of HOXA9-bound peaks in SEM cells.

Page 6 lines 123-126 (Supplementary Figure S2C); page 11 lines 227-232: The authors pick several sgRNA to validate by CPA – but only give information about the target loci for some of them (e.g. no information on sg6931). Where are these HOXA9 peaks (are they at promoters or distal loci)? Do they look like enhancers? These are referred to as being HOXA9-bound noncoding regions, but there’s no evidence provided to support this assertion. It would be nice to see a couple of (zoomed in) examples of where the sgRNA binds relative to the HOXA9 peak (and consensus motif), for example for the sgRNAs targeting the FLT3 enhancer.

Response: We thank the reviewer for this comment. Among the validated candidates, we have shown the data to support the HOXA9-bound peaks at the distal region of FLT3 and intronic region of CDK6 overlap with H3K27ac ChIP-seq peaks. These regions were also looped to the corresponding gene promoters based on the HiC data. In addition, we provided more validated peaks targeting AHI1, DCAF11, PEBP4, and ZCHC7 that also overlapped with enhancer marker H3K27ac in SEM cells.

New Figure 3A. Enhancer profiling of HOXA9-bound site in FLT3 enhancer by H3K27ac ChIP-seq in SEM.

New Supplementary Figure S7. Enhancer profiling of HOXA9-bound sites by H3K27ac and BRD4 ChIP-seq in SEM.

Page 8 lines 167-169: The big piece of data that's essential for arguing that h-FLT3 is important for upregulating FLT3 expression is to mutate the locus and assess its effect on FLT3 expression – but you don't do this until Figure 3, after you've already concluded the locus is important! This seems like an odd order in which to present this.

Response: We thank the reviewer for this comment. We apologize for the confusion and have revised the logic flow. We identify the top sgRNA hits at the distal enhancer of FLT3, a putative enhancer active in MLL-r leukemia. However, we did not know if the big chunk of enhancer and activity relies on HOXA-binding. Therefore, we used genome-editing tools such as CRISPR/Cas9 and dCas9-KRAB to target the binding site of HOXA9, leading to reduced cell fitness.

4. Considering how much the conclusions of the paper are dependent on the sgRNA-targeted mutation of HOXA9 binding sites, it's surprising that there is little or no validation that HOXA9 binding is lost in the transduced cells. I recognize that ChIP for HOXA9 is extremely difficult, but proxy assays could be used – for example, ChIP showing a reduction in H3K27ac, or ATAC-seq, at targeted enhancer loci (such as h-FLT3). If HOXA9 binding is functionally important, its loss should have effects on local chromatin.

Response: We thank the reviewer for this comment. We agree with the reviewer that endogenous HOXA9 ChIP-seq is the best approach to confirm occupancy loss in the edited cells. However, as the reviewer also pointed out, the endogenous HOXA9 ChIP-seq is very difficult. Therefore, as suggested by the reviewer, we conducted ATAC-seq in FLT3-DE1 and CDK6 intron-targeted cells, leading to a significant chromatin accessibility decrease.

New Figure 4C. Chromatin accessibility profiling in the HOXA9-bound site in the distal enhancer of FLT3 upon CRISPRi

New Figure 6E. Characterization of the chromatin conformation change upon dCas9-KRAB targeting against the HOXA9-bound site in the intron of CDK6. HiC, HOXA9 ChIP-seq, H3K27ac ChIP-seq, and BRD4 ChIP-seq tracks were shown to characterize the epigenetic status of the HOXA9-bound site.

As highlighted by the authors in several places, these HOXA9-bound peaks are also bound by a number of other transcription factors. Therefore, the CRISPR dropout screen can't be used to directly ascribe a functional role to HOXA9 binding at target loci. The authors can say that these DNA elements are important to growth/survival, but they can't say that HOXA9 binding specifically is important, as it may be one or more of the other TFs also seen to bind at them.

Response: We thank the reviewer for this comment. It is excellent that other co-factors may bind to the same target loci with HOXA9. Indeed, we observed other TF occupancy in the same HOXA9-bound regions. We will be careful with the tone to describe the HOXA9 binding and function. We are very interested in studying these co-factors in the future.

New Figure 3B. Transcription factor binding profiling of HOXA9-bound site in FLT3 distal enhancer.

5. There are several instances of the same data being shown twice in the paper (or virtually the same experiment being conducted twice, with equivalent data being shown). This is particularly prevalent for the growth competition assays. For example, Figure 3C: how is this experiment different from Supplementary Figure S2C? These are both CPA time courses in SEM cells targeting h-FLT3. Figure 4H appears to be the same data as Supplementary Figure S4A. Figure 6H: This is the same as Supplementary Figure S2C (for SEM cells). The authors also target the same loci in MOLM13 cells, but don't explain the relevance of this, or comment on the result. There is an important point about these effects not being specific to SEM cells, but the authors need to make it.

Response: We thank the reviewer for this comment. We apologize for the error. We have deleted this repeated data and explained more detail in the corresponding context.

Figure 5B/C: There is no need to repeat the same data in these figures as a pool as well as separately – if the authors really want to do this, they should move one set of the bar charts into supplementary.

Response: We thank the reviewer for this comment. We apologize for the confusion. We have now carefully revised the manuscript as suggested.

6. Figure 2C/D: It is very hard to interpret this Hi-C/HiChIP data – please annotate to make it clearer what you are trying to show (i.e. the interaction between the FLT3 promoter and enhancer). The resolution of these techniques is not ideal for exploring how CTCF degradation affects the interaction between the HOXA9 binding site and FLT3 promoter. It would be more informative to look at the effect

of CTCF degradation by Capture-C. Capture-C shows that the region interacting with the promoter is much broader than just the CTCF peak (without annotation it's hard to see where the CTCF peak lies). Does the entirety of this interaction disappear when upon CTCF degradation, or just the interaction at the CTCF peak? This may help to explain why CTCF degradation does not affect FLT3 expression. This would also complement the Capture-C experiment with the sgFLT3-DE1/2 dCas9, by demonstrating whether all or only some of the interactions are dependent on CTCF.

Response: We thank the reviewer for this comment. We apologize for the confusion. We have revised this data by only showing HiC and provided chromatin looping information upon CTCF degradation. Interestingly, upon acute CTCF degradation, chromatin looping between h-FLT3 and FLT3 promoter maintains the same, suggesting other looping factors may play essential roles.

New Figure 2. (A) schematic diagram of auxin-inducible degron (AID) system to target and acutely deplete CTCF protein. (B) Immunoblotting was conducted to confirm the acute protein degradation of CTCF in the presence of auxin for 24 hours. Immunoblotting was also performed to detect the expression of FLT3 protein upon acute degradation of CTCF. GAPDH was included as an internal control. (C) Q-PCR was used

to examine the mRNA expression of FLT3 upon CTCF protein degradation. Data are shown as mean values \pm SEM of three biological replicates (center of the error bar). P values were estimated using a two-tail t-test. (D) Capture-C and ATAC-seq were conducted to characterize the chromatin accessibility change and chromatin conformation change, respectively. The biotin-labeled DNA oligos against the FLT3 promoter were used as baits to quantify chromatin looping between the FLT3 promoter and the h-FLT3 enhancer.

7. Figure 6E: The authors need to make it clear that this is pathway analysis of the overlap DEGs that are affected both by Dox and 5-Ph-IAA treatment. I'm not sure what the logic is for only looking at genes that are sensitive to both HOXA9 overexpression and loss of that overexpression? This seems like a very complicated experiment to do rather than just knocking down HOXA9 (which would more directly test a requirement for HOXA9). Did the authors try this?

Response: We thank the reviewer for this comment. The HOXA9-miniAID cassette was ectopically expressed in the wildtype SEM and MOLM13 cells. The endogenous HOXA9 expression is still active in these engineered cells. Therefore, we cautiously move this piece of data in the revised manuscript.

Figure 6G: Overexpression of HOXA9 results in strong downregulation of XBP1, but sg3159 (which presumably disrupts HOXA9 binding) also downregulates XBP1 expression. On the face of it, these results are counterintuitive – how do the authors explain this? This argues that the role of HOXA9 is more complex than presented, and the authors should discuss possible interpretations of the data.

Response: We thank the reviewer for this comment. The HOXA9-miniAID cassette was ectopically expressed in the wildtype SEM and MOLM13 cells. The endogenous HOXA9 expression is still active in these engineered cells. Therefore, we cautiously removed this piece of data in the revised manuscript.

8. Figure 3H: OCI-AML2 cells appear to be less sensitive to sgRPS19 as well as sgFLT3-DE-1/2, suggesting that these cells may be more resistant to editing in general. If the authors wish to conclude that these cells are less dependent on the h-FLT3 enhancer, they should sequence the targeted locus in the transduced cells to demonstrate that the region has been successfully mutated.

Response: We thank the reviewer for this comment. We have sequenced the targeted pool population of OCI-AML2 cells. The indel frequency is more than 90% in the target regions.

Supplementary Figure S6C: sgJUN1/2 only very weakly phenocopy sg2571 in the CPA. This suggests that this either the sgRNA sequences targeting JUN are ineffective (which can be assessed by gDNA

sequencing and/or western blotting for JUN) or that the sg2571 locus has additional/alternative regulatory functions. Please discuss.

Response: We thank the reviewer for this comment. Given the structure of the revised manuscript was changed, we have deleted this piece of data to avoid any confusion.

Page 11 lines 249-250: “among the nearby genes of another candidate sgRNA (sg3243), only expression of the BAHCC1 gene was decreased by Cas9-sg3243”. Where are the data to support this statement? I’m not sure why the authors show the different TFs bound at the BAHCC1 locus (Figure 7A) as they don’t make any reference to this in the text.

Response: We thank the reviewer for this comment. Given the structure of the revised manuscript was changed, we have deleted this piece of data to avoid any confusion.

9. Figure 8 is entirely speculative and barely discussed at all in the text. The suggestion in the figure that deletion of a HOXA9 binding site would produce a different phenotype to HOXA9 knock-out is an interesting one – but nowhere is this discussed. Of course, an alternative explanation (not addressed in the paper) is that mutation of the sequence disrupts the binding of multiple TFs (not just HOXA9), and perhaps one or more of these TFs has a (more?) important role in gene activation.

Response: We thank the reviewer for this comment. We apologize for the confusion. We now lower the tone and discuss more about this model and other possible mechanisms.

Minor points:

There are a lot of typographical and grammatical errors in this paper which need to be addressed. In several cases the authors refer to several figure panels in one go, where it would be much easier for the reader to follow if individual panels were used to make specific points.

Response: We apologize for the errors and have carefully revised the manuscript.

1. Supplementary Figure S1D: what is the difference between the motif enrichment on the left and the right? Are these called using different algorithms? There’s no detail in the legend. Why does this figure refer to 3774 HOXA9 peaks and 2820 MEIS1 peaks, when the text refers to 1806 HOXA9 and 6725 MEIS1 peaks?

Response: We thank the reviewer for this comment. Given the structure of the revised manuscript was changed, we have deleted this piece of data to avoid any confusion.

2. Page 6 line 112: “a known distal enhancer of the FLT3 locus” needs a citation.

Response: New references were added.

3. Page 6 line 115 (Supplementary Fig S2A): unless I have misunderstood (in which case you need to

explain this more clearly) this is not motif analysis. Please rephrase. What is your logic for looking at these transcription factors in particular?

Response: We thank the reviewer for this comment. These are known TFs essential for leukemia cell maintenance.

4. Page 7 lines 138-139: needs proof-reading.

Response: We have made corrections.

5. Page 7 line 145: you need to add “in SEM cells (Figure 2B).” You are using data from multiple MLLr cell lines here interchangeably – I think this is justified, but you need to make sure you are explicit about which dataset you are referring to at each point.

Response: We thank the reviewer for this comment. We have added the GEO number of each dataset to avoid any confusion.

6. Page 8 lines 160-169: you should refer to individual panels of Supplementary Figure S4 to support specific statements – not just cite them all together at the end. This makes it much harder for the reader to assess the data.

Response: We thank the reviewer for this comment. We made changes as requested to avoid any confusion.

7. Page 8 line 162: “dependency” should be “dependent”.

Response: We thank the reviewer for this comment. We made corrections.

8. Page 11 line 239: “deferentially” should be “differentially”.

Response: We thank the reviewer for this comment. We made corrections.

9. Page 11 lines 246-247: “We could match the sg3159 editing effect with transcriptionally affected genes located in the same topological associated domain (Figure 6G).” I think there must be a mistake in this sentence, because it doesn’t match the conclusion from Figure 6G.

Response: We thank the reviewer for this comment. Given the structure of the revised manuscript was changed, we have deleted this piece of data to avoid any confusion.

10. Page 13 lines 297-298: needs proof-reading.

Response: We thank the reviewer for this comment. We made corrections.

11. Page 16 line 355: “illumine” should read “Illumina”.

Response: We thank the reviewer for this comment. We made corrections.

12. Page 26 lines 663-664: needs proof-reading.

Response: We thank the reviewer for this comment. We made corrections.

13. Pages 26-27: Legend is missing for Fig2D.

Response: We thank the reviewer for this comment. We made corrections.

14. Figure 6B: The labels for the bands on the western need to be more clearly aligned (> symbols are not ideal) – for the lower blot it's unclear which band the arrow is pointing at.

Response: We thank the reviewer for this comment. Given the structure of the revised manuscript was changed, we have deleted this piece of data to avoid any confusion.

15. Supplementary Figure S3A: legend should read “Depletion of CTCF specifically decreased the chromatin accessibility at the CTCF binding site of the h-FLT3 enhancer”. Use of the term “target site” is ambiguous (do you mean the HOXA9 binding site, or sgRNA target site?).

Response: We thank the reviewer for this comment. We made corrections. “Target site” is the sgRNA-target site.

16. Supplementary Figure S6A: This is referred to in the text in the context of sg2571/JUN, but the figure appears to be about sg8105.

Response: We thank the reviewer for this comment. Given the structure of the revised manuscript was changed, we have deleted this piece of data to avoid any confusion.

Reviewer #3, expertise in CRISPR screens for haematological malignancies (Remarks to the Author):

In the manuscript entitled "Comprehensive and functional interrogation of the HOXA9 downstream regulation mechanism in MLL-r leukemia", Wright A & Zhao X et al. provided relevant data on the downstream functional effectors of HOXA9 which could represent potential therapeutic targets for MLL-r leukemias.

Major comments:

- Authors should explain which cell lines they reanalyzed using CHIP assays. In the results, they only named B-ALL SEM and "other cell lines".

Response: We thank the reviewer for this great comment. The reviewer pointed out that our noncoding CRISPR screens against HOXA9's targets rely on an "unpublished" and GEO deposit of a HOXA9 ChIP-seq dataset using MLL-r SEM cells. The quality of this ChIP-seq dataset is good based on motif analysis. However, a significant variation was also observed from two and only replicates. Therefore, to address this question, we have spent enormous efforts to set up the ChIP-seq assay using MLL-r SEM cells in our lab. In brief, to cope with the difficulty of ChIP-seq against endogenous HOXA9 by lacking good antibodies, we established the inducible TetOn system to express HOXA9-HA cDNA upon doxycycline treatment. The no-drug treatment groups show no leaking and serve as a perfect isogenic control. By utilizing this new model system, we have successfully conducted a reproducible ChIP-seq (three replicates for dox-treatment and three no-drug treatments). We have identified 229 high-quality HOXA9-binding peaks, including the most significant peak at the distal enhancer of *FLT3* (FLT3 DE1), which we identified in the first submission. Given the peak number variation between our ChIP-seq and the publicly available one (229 vs. 1,806), we generated a new noncoding CRISPR library against all 229 HOXA9-binding peaks and then conducted a new dropout CRISPR screen. To our understanding, this is the best ChIP-seq data available so far. We systematically validated the top candidates in the following study by competitive proliferation assay, chromatin conformation characterization and chromatin accessibility profiling analysis. These efforts solidify our previous efforts to identify HOXA9's downstream targets in the MLL-r leukemia setting.

New Figure 1. (A) Diagram of the all-in-one inducible TetOn system to ectopically express HOXA9-HA cDNA. (B) Genomic distribution of the 229 HOXA9 ChIP-seq peaks. (C) The top three consensus motifs were shown using the motif analysis algorithm against 229 peaks compared with control peaks. (D) Heat maps of HOXA9-HA ChIP-seq peaks compared with publicly available HOXA9 and MEIS1 ChIP-seq datasets and additional transcription factor ChIP-seq datasets in SEM cells.

- Could the authors further explain and validate the differences among common binding peaks of HOXA9 and MEIS1, as well as those that are specific for HOXA9?

Response: We thank the reviewer for this great comment. Our previous noncoding CRISPR screen against HOXA9's targets relied on an "unpublished" and GEO deposit of a HOXA9 ChIP-seq dataset using MLL-r SEM cells. The quality of this ChIP-seq dataset is good based on motif analysis. However, a significant variation was also observed from two and only replicates. It is very hard to conclude the binding peak differences between HOXA9 and MEIS1 are due to the biological context or technical variation. Also, the motif analysis suggests that MEIS1 and HOXA9-bound peaks are similar.

- How did you define the subgroups of AML patients according to FLT3 expression (Figure S4D)?

Response: We thank the reviewer for this great comment. The TCGA AML survival data is from the GEPIA2 database (<http://gepia2.cancer-pku.cn>). The default algorithm unbiasedly defines the low and high populations by equal patient numbers.

- The authors should explain with more details the approach to OE and KD genes (Figure 6A).

Response: We thank the reviewer for this comment. The HOXA9-miniAID cassette was ectopically expressed in the wildtype SEM and MOLM13 cells. The endogenous HOXA9 expression is still active in these engineered cells. Therefore, we cautiously removed this piece of data in the revised manuscript.

- BAHCC1 is a potential target that should be validated in an in vivo model.

Response: We thank the reviewer for this comment. BAHCC1 was recently reported to bind H3K27me3 via a conserved BAH module to mediate gene silencing and oncogenesis (Nat Genet. 2020 Dec;52(12):1384-1396.). They have already systematically conducted in vivo experiments in primary leukemia. Depletion of BAHCC1, or disruption of the BAHCC1BAH-H3K27me3 interaction, causes de-repression of H3K27me3-targeted genes involved in tumor suppression and cell differentiation, leading to suppression of oncogenesis. Introducing a germline mutation at Bahcc1 to disrupt its H3K27me3 engagement in mice causes partial postnatal lethality. Instead, we conducted more validation work on other new candidates from our new noncoding CRISPR. In addition to FLT3, we identified a novel functional HOXA9-binding peak in the intronic region of CDK6, which is a plausible intronic enhancer looped to the CDK6 promoter in MLL-r leukemia cells. Targeting this peak by CRISPR/Cas9 or dCas9-KRAB significantly reduced chromatin accessibility and leukemia cell fitness. These efforts suggest that targeting CDK6 by FDA-approved CDK4/6 inhibitors could potentially benefit the MLL-r leukemia therapy alone or synergy with other drugs. Although the current scope of our study is not directly focusing on preclinical and clinical treatment, we are very interested in testing these hypotheses in future research.

- The authors used a cohort of 106 AML patients from TCGA. However, these results should be validated in other cohorts.

Response: We thank the reviewer for this comment. We now provide the survival data from the TCGA AML cohort, which indicates that the high expression of HOXA9 is associated with poor survival (A), similar to the observation for FLT3. TARGET and BEAT cohorts (B and C) also observed a similar trend. We did not detect significant correlations from other validated candidates. These data were shown as **New Supplementary Figure S2**.

Reviewers' Comments:

Reviewer #1:

Remarks to the Author:

The authors have clarified all the questions I raised in my previous review. The revision is satisfactory.

Reviewer #2:

Remarks to the Author:

This paper has been substantially rewritten and is much improved, with a significant body of new experiments including validation of additional HOXA9 binding sites. In addition, the bioinformatic analyses conducted are more clearly explained. Overall, I am satisfied that the authors have addressed my concerns.

However, I feel that some of the assertions/conclusions made in the discussion section are not supported by the evidence presented, and I would suggest that these be toned down/reworded. In particular:

1. Supplementary Figures 6 and 7 contain data and as such should be referred to first in the results section, not the discussion.

2. Lines 254-259: What is the evidence that chromatin accessibility is the major mechanism controlling enhancer activity of HOXA9-bound sites? This is correlative, not necessarily causative. It would be more accurate to say that these HOXA9 binding sites are required for enhancer accessibility at the CDK6 and FLT3 loci (but not at RUNX1 and DCAF11). (NB the label in Supplementary Fig 6D needs changing from sgDCAF to sgRUNX1 target.)

3. Lines 265-268: The data presented here do not show that chromatin-looping mediated enhancer regulation is "critical" (ie required for activity) – they simply show that the FLT3 and CDK6 enhancers interact with the target gene promoters. In fact, the Capture-C at FLT3 conducted following sgFLT3 enhancer deletion show no change in interaction frequency (Fig 4g), so this does not provide evidence that the interaction is required (or dependent on the presence of the targeted sequence). It is likely that this looping is important for enhancer activation of the target gene, but no evidence is provided here to show essentiality.

Reviewer #3:

Remarks to the Author:

In the revised manuscript entitled "Systematic characterization of the HOXA9 downstream targets in MLL-r leukemia by noncoding CRISPR screens", Wright A & Zhao X et al. have provided more relevant data on the downstream functional effectors of HOXA9 and they have addressed most of my previous comments. However, since they have included novel results and data, I have several concerns in this new version of the manuscript:

Major comments

- The authors should give more details in Figure Legends. For example, in the Fig Suppl 1.B, they should explain what represents the intensity of the orange color and numbers in the bottom of the heatmap. Also, in figure legends of Figure 4H-J, authors should provide more information separately.
- Could the authors explain in detail how they perform HOXA9-HA ChIP in the methodology section?
- In this version, it is not clear the overlapping of the publicly available HOXA9 and HOXA9-HA ChIP. Could the authors clarify this in the manuscript? In Supp Figure 1B, authors show 1806 peaks but in the Figure 1D the authors show 229.

- I miss the statistical analysis in Figure 4H.
- Authors showed new results related to CDK6 and HOXA9. Is this target specific in MLL-r background? Authors should complement their results with other non-MLL-r cell lines to see the effect of KO or KD CDK6 in these cell lines. Also, it will be interesting to include further validation such as an in vivo CPA experiment and the effects of CDK6 inhibitors in cell growth.

Minor comments:

- In the Abstract section, authors should review the verb tenses (for example, the third sentence).
- There is a typo in the sentence: "ATAC-sq peak files were provided in Supplementary Table S5".

REVIEWER COMMENTS (2nd round review)

Reviewer #1, The authors have clarified all the questions I raised in my previous review. The revision is satisfactory.

Response: Thank you for recognizing the value of our efforts in the revision.

Reviewer #2, This paper has been substantially rewritten and is much improved, with a significant body of new experiments including validation of additional HOXA9 binding sites. In addition, the bioinformatic analyses conducted are more clearly explained. Overall, I am satisfied that the authors have addressed my concerns.

Response: Thank you for recognizing the value of our efforts in the revision.

However, I feel that some of the assertions/conclusions made in the discussion section are not supported by the evidence presented, and I would suggest that these be toned down/reworded. In particular:

1. Supplementary Figures 6 and 7 contain data and as such should be referred to first in the results section, not the discussion.

Response: We thank the reviewer for this comment. We have referred some data to the corresponding result sections (page 11).

2. Lines 254-259: What is the evidence that chromatin accessibility is the major mechanism controlling enhancer activity of HOXA9-bound sites? This is correlative, not necessarily causative. It would be more accurate to say that these HOXA9 binding sites are required for enhancer accessibility at the CDK6 and FLT3 loci (but not at RUNX1 and DCAF11). (NB the label in Supplementary Fig 6D needs changing from sgDCAF to sgRUNX1 target.)

Response: We thank the reviewer for this comment. Given the primary effect of dCas9-KRAB editing on chromatin is to hetero-chromatinize the site and recruit H3K9me3 modification, we concluded that chromatin accessibility is the major mechanism controlling enhancer activity of HOXA9-bound sites. We agree with the reviewer that it would not be accurate, given other potential mechanisms may also contribute to the observation. Therefore, we changed the description to say that “these HOXA9 binding sites are required for enhancer accessibility at the CDK6 and FLT3 loci (but not at RUNX1 and DCAF11)” as suggested by the reviewer (page 12). We also apologize for the labeling error. We have changed the label in Supplementary Fig 6D from sgDCAF11 to sgRUNX1 target.

3. Lines 265-268: The data presented here do not show that chromatin-looping mediated enhancer regulation is “critical” (ie required for activity) – they simply show that the FLT3 and CDK6 enhancers interact with the target gene promoters. In fact, the Capture-C at FLT3 conducted following sgFLT3 enhancer deletion show no change in interaction frequency (Fig 4g), so this does not provide evidence that the interaction is required (or dependent on the presence of the targeted sequence). It is likely that this looping is important for enhancer activation of the target gene, but no evidence is provided here to show essentiality.

Response: We thank the reviewer for this comment. To explain the observation that CRISPR deletion and CRISPRi targeting both decreased FLT3 expression but not the promoter/enhancer interaction frequency, we hypothesized that the HOXA9-bound site is responsible for the enhancer activity maintenance, while

there is another looping factor (not CTCF) bound in a separate site accountable for the chromatin looping. We will continue to study the CTCF-independent chromatin looping maintenance in the future.

Reviewer #3, In the revised manuscript entitled "Systematic characterization of the HOXA9 downstream targets in MLL-r leukemia by noncoding CRISPR screens", Wright A & Zhao X et al. have provided more relevant data on the downstream functional effectors of HOXA9 and they have addressed most of my previous comments.

Response: Thank you for recognizing the value of our efforts in the revision.

However, since they have included novel results and data, I have several concerns in this new version of the manuscript:

Major comments

- The authors should give more details in Figure Legends. For example, in the Fig Suppl 1.B, they should explain what represents the intensity of the orange color and numbers in the bottom of the heatmap. Also, in figure legends of Figure 4H-J, authors should provide more information separately.

Response: We thank the reviewer for this comment. The color indicates the peak counts. Based on the peak counts per million, we normalize the signal of each HOXA9-bound peak to 15 million in total. The darker the color stands for higher peak intensity, and the lighter the less intensity. We also added more description about Figures 4H-4J, including DepMap analysis and drug response against Gilteritinib (FLT3 inhibitor) treatment. These data suggest higher *FLT3* expression confers more sensitivity against Gilteritinib treatment. When the HOXA9-bound site was targeted, the *FLT3* expression decreased, indicating resistance to Gilteritinib treatment as expected.

- Could the authors explain in detail how they perform HOXA9-HA ChIP in the methodology section?

Response: We thank the reviewer for this comment. We have now provided more experimental details regarding our HOXA9-HA ChIP. To cope with the common problem and difficulty of ChIP-seq against endogenous HOXA9 by lacking suitable antibodies, we established the inducible TetOn system to express HOXA9-HA cDNA upon doxycycline treatment in MLL-r SEM cells. Twenty million HOXA9-HA cells were treated with 1 μ g/mL doxycycline for 48 hours to induce the HOXA9-HA protein expression. DMSO treatment was used as a negative control. Cells were fixed with 1% formaldehyde for 5 mins at room temperature (Covaris TruChIP Chromatin Shearing Kit). Nuclei were prepared according to the TruChIP protocol and chromatin was sheared in a Covaris milli tube using the Covaris M220 ultrasonicator set at a duty factor of 10 and 200 cycles/burst for 10 min at set point 6 °C. Sheared chromatin was centrifuged for 10 mins at 8000 \times g and clarified chromatin was moved to a new 1.5-mL Eppendorf tube. Chromatin was amended to a final concentration of 50 mM Tris-HCL pH 7.4, 100 mM NaCl, 1 mM EDTA, 1% NP-40, 0.1% SDS, and 0.5% Na deoxycholate plus protease inhibitors. About 60 μ l of washed anti-HA magnetic beads (Pierce, catalog #88837) were added to the chromatin overnight with rotation at 4 °C. The next day, samples were placed on a magnetic stand, unbound chromatin was removed, and beads were washed 2 times with wash buffer 1 (50 mM Tris-HCL pH 7.4, 1 M NaCl, 1 mM EDTA, 1% NP-40, 0.1% SDS, 0.5% Na deoxycholate plus protease inhibitor) followed with wash buffer 2 (20 mM Tris-HCL pH 7.4, 10 mM MgCl₂, 0.2% Tween-20 plus protease inhibitor). The beads were resuspended in wash buffer 2 and transferred to a new 1.5-mL Eppendorf tube. Samples were placed on a magnetic stand to remove the wash buffer. DNA was eluted and de-crosslinked in 1X TE plus 1% SDS, proteinase K, and 400 mM NaCl at 65 °C for 4 h. DNA

was precipitated by phenol, chloroform, and isopropyl alcohol. Libraries were constructed by NEBNext Ultra II NEB Library Prep Kit and NEBNext Multiplex oligos for Illumina. From the three replicates for dox-treatment and three no-drug treatments, we have identified 229 high-quality HOXA9-binding peaks.

- In this version, it is not clear the overlapping of the publicly available HOXA9 and HOXA9-HA ChIP. Could the authors clarify this in the manuscript? In Supp Figure 1B, authors show 1806 peaks but in the Figure 1D the authors show 229.

Response: We thank the reviewer for this comment. The only publicly available HOXA9 ChIP-seq data conducted in MLL-r SEM cells were deposited in GEO (GSE38339) without further peer-review publication. Given the peak number variation between the two replicates in their study, we can only lower the statistical criteria to call reproducible peaks (FDR<0.05 in one sample and FDR<0.5 in the other). Based on this analysis, we identified 1,806 HOXA9-bound peaks in their study. In our research, we have spent enormous efforts to set up the ChIP-seq assay using MLL-r SEM cells in our lab. We established the inducible TetOn system to express HOXA9-HA cDNA upon doxycycline treatment. The no-drug treatment groups show no leaking and serve as a perfect isogenic control. By utilizing this new model system, we have successfully conducted a reproducible ChIP-seq (three replicates for dox-treatment and three no-drug treatments). We have identified 229 high-quality HOXA9-binding peaks. Therefore, our analysis stringency is much higher than the publicly available dataset. Given the difference in statistical rigor and peak calling strategy, we are surprised that only 91 HOXA9-bound peaks overlap between 229 and 1,806 (one peak may overlap with several in another dataset). Our study only focused on the functional screen against our 229 peaks, as shown in Figure 1D. To indicate the variation of peaks between their replicates, we also use 1,806 peaks as a reference to align the signals with our data. The data suggest that more peaks show relatively stronger signals in our Dox-treatment groups than in the no-Dox setting. However, they did not pass the stringent statistical test in our study.

- I miss the statistical analysis in Figure 4H.

Response: We apologize for the confusion. The gene expression and dependency score of selective cell lines were collected from the DepMap database and used to indicate the variation of FLT3 dependency in leukemia cell lines. The highlighted cell lines were used for the Gilteritinib drug response in Figures 4I and 4J. We do not intend to claim statistical differences within the cell lines, given that other factors may also affect the Gilteritinib drug response.

- Authors showed new results related to CDK6 and HOXA9. Is this target specific in MLL-r background? Authors should complement their results with other non-MLL-r cell lines to see the effect of KO or KD

CDK6 in these cell lines. Also, it will be interesting to include further validation such as an *in vivo* CPA experiment and the effects of CDK6 inhibitors in cell growth.

Response: We thank the reviewer for this comment. We agree with the reviewer that CDK6 is an exciting target, and it will be good to include further validation such as an *in vivo* CPA experiment and the effects of CDK6 inhibitors in cell growth. However, the *in vivo* CPA experiment requires at least several months. Therefore, we conducted a CDK6 inhibitor treatment experiment in MLL-r and non-MLL-r acute lymphoblastic leukemia (ALL) cell lines. Given only one B-ALL MLL-r ALL line (SEM) is available on our hand, we did not see selective sensitivity of CDK4/6 inhibitors. In contrast, the control FLT3 inhibitor Quizartinib shows lower IC50 in SEM cells than other ALL lines (Nalm6, 697, REH, RPMI8402, and MOLT4).

Next, we also conducted CDK4/6i inhibitor (Ribociclib) treatment and the MTT assay on human MLL-r AML cell lines MOLM13, OCI-AML2, MV4,11 and non-MLL-r AML cell line K562. Compared to the only one non-MLL-r AML cell line K562, the other three MLL-r AML cell lines and B-ALL SEM cell line demonstrated notable increased sensitivity towards Ribociclib. However, we want to point out that the three CDK4/6 inhibitors (Abemaciclib, Palbociclib, and Ribociclib) target both CDK4 and CDK6 and possibly other CDKs. However, CRISPRi and CRISPR disruption only precisely target regulator elements of CDK6. Moreover, we also need to conduct the drug treatment on more AML and ALL cell lines to make a solid conclusion. We will continue working on *in vivo* testing and revealing potential clinical applications in future studies.

Minor comments:

- In the Abstract section, authors should review the verb tenses (for example, the third sentence).

Response: We thank the reviewer for this comment. We fixed the grammar and spelling errors.

- There is a typo in the sentence: "ATAC-sq peak files were provided in Supplementary Table S5".

Response: We thank the reviewer for this comment. We have made corrections to "ATAC-seq."

Reviewers' Comments:

Reviewer #2:

Remarks to the Author:

I am satisfied that the reviewers have addressed my comments.

Reviewer #3:

Remarks to the Author:

The authors have clarified most of the questions I raised in my previous review.

Minor comments

- Regarding figures 4I-J, I was wondering if the authors could calculate the statistics. For example in Figure J, the authors could calculate if there are differences between sgNT versus sgFLT3; in Figure I, it would be interesting if there are differences between MLL-r vs non-MLL-r.
- In order to validate that CDK6 is a specific target for MLL-r leukemias, the authors could incorporate the results in MLL-r and non-MLL-r leukemias and inhibition of CDK6. Also, it would be interesting to validate these results when CDK6 is knockdown (maybe b CRISPR or sh/siRNA)

REVIEWER COMMENTS (3rd round review, version B)

Reviewer #2: I am satisfied that the reviewers have addressed my comments.

Response: Thank you for recognizing the value of our efforts in the revision.

Reviewer #3, The authors have clarified most of the questions I raised in my previous review.

Minor comments

- Regarding figures 4I-J, I was wondering if the authors could calculate the statistics. For example, in Figure J, the authors could calculate if there are differences between sgNT versus sgFLT3; in Figure I, it would be interesting if there are differences between MLL-r vs non-MLL-r.

Response: We thank the reviewer for this comment. Given that only a limited number of MLL-r and non-MLL-r B-ALL cell lines were used in this panel, we feel it will be more appropriate to conclude the differences between MLL-r vs non-MLL-r from the statistical perspective with more cell line data. However, there are no additional well-characterized MLL-r B-ALL cell lines available yet. We will continue to work in this direction in the future.

- In order to validate that CDK6 is a specific target for MLL-r leukemias, the authors could incorporate the results in MLL-r and non-MLL-r leukemias and inhibition of CDK6. Also, it would be interesting to validate these results when CDK6 is knockdown (maybe b CRISPR or sh/siRNA)

Response: We thank the reviewer for this comment. We agree with the reviewer that CDK6 is an exciting target. However, we do not think CDK6 is a specific target only for MLL-r leukemias. As this expert reviewer is aware, CDK6 is a pan-cancer target. Our study revealed a possible mechanism that HOXA9 could directly regulate CDK6, which adds an additional regulation axis in MLL-r leukemia. However, given only one B-ALL MLL-r ALL line (SEM) is available on our hand, we did not see selective sensitivity of CDK4/6 inhibitors as we have shown in the last round of revision. Moreover, we realize it will have to conduct CRISPR or sh/siRNA experiments in more AML and ALL cell lines to make a solid conclusion, which we haven't set up in the lab. We will continue working on *in vivo* testing and revealing potential clinical applications in future studies.